# Towards a Unified View of Large Language Model Post-Training

## Abstract

Many approaches with seemingly disparate losses exist for post-training modern language models, such as Reinforcement Learning (RL) and Supervised Fine-Tuning (SFT). In this paper, we show that these approaches are not in contradiction, but are instances of a single optimization process. We derive the Unified Policy Gradient Estimator (UPGE), a framework with four interchangeable parts that unifies a wide spectrum of post-training approaches through their loss gradient form. We further present the calculations of these methods as the gradient of a common objective under different data distribution assumptions and various bias-variance tradeoffs. Motivated by our theoretical findings, we propose Hybrid Post-Training (HPT), an algorithm that dynamically selects different training signals. HPT is designed to yield both effective exploitation of demonstration and stable exploration without sacrificing learned reasoning patterns. We provide extensive experiments and ablation studies to verify effectiveness of HPT. Across six mathematical reasoning benchmarks and two out-of-distribution tasks, HPT consistently surpasses strong baselines across models of varying scales and families.

## 1 Introduction

Reinforcement Learning has played an integral role in enhancing the reasoning capabilities of large language models (LLMs) (Jaech et al., 2024; Team et al., 2025; Guo et al., 2025). However, applying RL directly to a base model (i.e., "Zero RL") (Zeng et al., 2025b) presupposes a certain level of inherent capability. This method often falters when applied to weaker models or tasks of high complexity, as the exploration process may fail to discover meaningful reward signals. Conversely, the classical Supervised Fine-Tuning (SFT) (Wei et al., 2021) offers a direct and efficient method to distill knowledge from high-quality, human-annotated data, but often curtails the model's exploratory capabilities. Consequently, a sequential "SFT-then-RL" pipeline (Yoshihara et al., 2025) has emerged as the standard, adopted by numerous state-of-the-art models. While effective, this multi-stage process is notoriously resource-intensive and usually requires careful tuning to ensure effectiveness.

To circumvent these challenges, recent works have focused on integrating SFT or SFT-style imitation learning losses directly with RL objectives (Yan et al., 2025; Fu et al., 2025; Zhang et al., 2025; Yu et al., 2025; Zeng et al., 2025a). In these approaches, the model is updated using a composite loss function. The balance between imitation and exploration components is governed by various strategies, including a fixed coefficient, a predefined schedule, a dynamic adjustment based on entropy, or a learnable parameter. Appendix A further discusses related work. These works predominantly treat SFT and RL losses as two distinct objectives. And a detailed analysis of *why these two learning signals can be effectively combined within a unified optimization process* remains largely unexplored.

Despite their distinct mathematical formulations, we find that the loss gradients of these approaches can be cast into a single form and jointly drive the optimization process under a unified framework. Inspired by Generalized Advantage Estimator (Schulman et al., 2015b), we introduce Unified Policy Gradient Estimator (UPGE), a framework that subsumes the gradients of various post-training objectives into one generalized expression with four interchangeable parts: *stabilization mask*, *reference policy*, *advantage estimate*, and *likelihood gradient*. We show that UPGE can be theoretically derived from a common objective, and the various forms of gradients are not conflicting but instead act as complementary learning signals that jointly guide optimization under this view. Further, through UPGE it becomes clear that these gradient estimators possess different characteristics, and there

Table 1: Theoretical unified view of various post-training algorithms of the large language model.

| Algorithm | Reference Policy | Advantage Estimate | Unified Policy Gradient Estimator |
|---|---|---|---|
| SFT | $\pi_{ref} = \pi_\theta$ | $\hat{A}_{SFT} \equiv 1$ | $\nabla\mathcal{J}_{SFT}(\theta) = \nabla\pi_\theta(\tau)\frac{\hat{A}_{SFT}=1}{\pi_\theta(\tau)}$ |
| **Online Reinforcement Learning Methods** | | | |
| PPO (Schulman et al., 2017) | $\pi_{ref} = \pi_{\theta_{old}}$ | $\hat{A}_{PPO}$ = GAE (Schulman et al., 2015b) | $\nabla\mathcal{J}_{PPO} = \nabla\pi_\theta(\tau)\frac{\hat{A}_{PPO}\mathbb{1}_{\text{Clip}}}{\pi_{ref}(\tau)}$ |
| GRPO (Shao et al., 2024) | $\pi_{ref} = \pi_{\theta_{old}}$ | $\hat{A}_{GRPO} = \frac{R(\tau_j)-\text{mean}(\{R(\tau_j)\}_{G_{on}})}{\text{std}(\{R(\tau_j)\}_{G_{on}})}$ | $\nabla\mathcal{J}_{GRPO} = \nabla\pi_\theta(\tau)\frac{\hat{A}_{GRPO}\mathbb{1}_{\text{Clip}}}{\pi_{ref}(\tau)}$ |
| REINFORCE (Ahmadian et al., 2024) | $\pi_{ref} = \pi_\theta$ | $\hat{A}_{REINFORCE} = \pm 1$ | $\nabla\mathcal{J}_{REF.}(\theta) = \nabla\pi_\theta(\tau)\frac{\hat{A}_{REF.}}{\pi_\theta(\tau)}$ |
| CISPO (Chen et al., 2025) | $\pi_{ref} = \pi_{\theta_{old}}$ | $\hat{A}_{CISPO} = \hat{A}_{GRPO}$ | $\nabla\mathcal{J}_{CISPO} = \nabla\pi_\theta(\tau)\frac{\hat{A}_{CISPO}\mathbb{1}_{\text{CIS-Mask}}}{\pi_{ref}(\tau)}$ |
| GSPO (Zheng et al., 2025) | $\pi_{ref} = \pi_\theta\left(\frac{\pi_{\theta_{old}}(\tau_{i,j}|q_i)}{\pi_\theta(\tau_{i,j}|q_i)}\right)^{1/|\tau_{i,j}|}$ | $\hat{A}_{GSPO} = \hat{A}_{GRPO}$ | $\nabla\mathcal{J}_{GSPO} = \nabla\pi_\theta(\tau)\frac{\hat{A}_{GSPO}\mathbb{1}_{\text{Seq-Clip}}}{\pi_{ref}(\tau)}$ |
| **Offline/Online Reinforcement Learning Methods** | | | |
| SRFT (Offline) (Fu et al., 2025) | $\pi_{ref} \equiv 1$ | $\hat{A}_{SRFT} = \frac{R(\tau_j)-\text{mean}(\{R(\tau_j)\}_{G_{on}\cup G_{off}})}{\text{std}(\{R(\tau_j)\}_{G_{on}\cup G_{off}})}$ | $\nabla\mathcal{J}_{SRFT} = \nabla\pi_\theta(\tau)\frac{\hat{A}_{SRFT}}{\pi_{ref}(\tau)=1}$ |
| LUFFY (Offline) (Yan et al., 2025) | $\pi_{ref} \equiv 1$ | $\hat{A}_{LUFFY} = \hat{A}_{SRFT}$ | $\nabla\mathcal{J}_{LUFFY} = \nabla\pi_\theta(\tau)\frac{\hat{A}_{LUFFY}}{\pi_{ref}(\tau)=1}f'_{\text{shape}}$ |

exists a bias–variance tradeoff in their respective gradient components, which suggests that an ideal post-training algorithm should dynamically select among them to suit different conditions. Building upon this unified perspective, we propose Hybrid Post-Training (HPT), a hybrid algorithm to dynamically choose more desirable training signals by switching between SFT and RL based on rollout accuracy feedback. This mechanism allows HPT to be intrinsically adaptive to models of varying capabilities and data of differing complexities.

Empirical evaluations demonstrate that HPT surpasses baselines such as SFT→GRPO and LUFFY on Qwen2.5-Math-7B, achieving a 7-point gain over our strongest baseline on AIME 24. Moreover, HPT yields substantial improvements even on smaller and weaker models, including Llama3.1-8B and Qwen2.5-Math-1.5B. Through detailed training dynamics and illustrative visualizations, we clearly reveal the features and underlying mechanisms of HPT. The following are several key takeaways:

> **Takeaways**
>
> 1. UPGE provides a theoretical unification of a wide spectrum of post-training algorithms, covering both SFT and RL losses within a single formulation (§ **2**).
>
> 2. HPT is capable of outperforming previous post-training and mixed-policy algorithms across a diverse range of models and benchmarks (§ **3**).
>
> 3. Dynamic integration of SFT and RL in HPT achieves the highest *Pass@1024*, facilitating enhanced exploration and generalization of the model (§ **4.1**).

## 2 A UNIFIED VIEW ON POST-TRAINING ALGORITHMS

Despite the substantial differences in loss formulations across various LLM post-training algorithms, we find that their loss gradients can be expressed in a single common form. We propose a unified framework for the gradient calculation, named Unified Policy Gradient Estimator (UPGE):

$$\text{grad}_{Uni} = \mathbb{1}_{stable}\frac{1}{\pi_{ref}}\hat{A}\nabla\pi_\theta.$$

In the following sections, we first provide a detailed analysis of the components of UPGE (§ 2.1). Then we theoretically derive the UPGE from a common objective (§ 2.2). Based on the unified perspective, we finally propose the Hybrid Post-Training (HPT) algorithm (§ 2.3).

### 2.1 COMPONENTS OF THE UNIFIED POLICY GRADIENT ESTIMATOR

We present the Unified Policy Gradient Estimator, our unified framework for gradient calculations. In Table 1, we list the UPGE instantiations for some representative post-training methods, with detailed

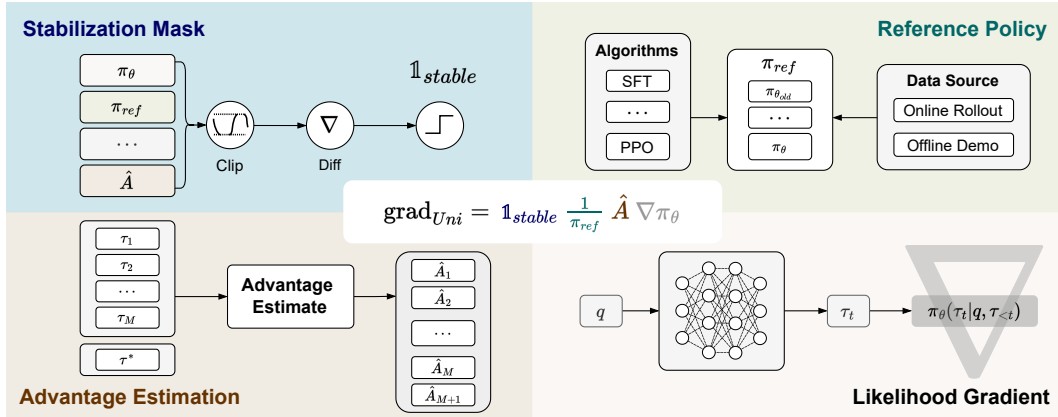

Figure 1: Illustration of the Unified Policy Gradient Estimator. The "$\nabla$" in the background of the Likelihood Gradient part refers to the calculation of the gradient with respect to the $\pi_\theta$.

derivations provided in Appendix B. UPGE comprises four components: *stabilization mask*, *reference policy*, *advantage estimate*, and *likelihood gradient*. We address each of the terms below.

**Stabilization Mask** $\mathbb{1}_{stable}$    Starting from PPO (Schulman et al., 2017), the stabilization mask was first derived as an approximation of the TRPO Algorithm (Schulman et al., 2015a). In practice, the PPO clipping addresses the instability issue during RL training by turning off the current gradient when the current iterate is considered unsafe. In consequent works in Table 1, many have provided their modifications on the stability mask, usually motivated by empirical evaluations.

**Reference Policy Denominator** $\pi_{ref}$    We note that our notion of reference policy on the denominator differs from the commonly used rollout policy $\pi_{\theta_{old}}$. This denominator denotes a token-level reweight coefficient, usually in the form of an inverse probability. There are multiple choices for this coefficient. For the case of SFT, the policy denominator uses the current policy $\pi_\theta(\tau)$, which is a result of $\mathcal{L} = -\log(\pi_\theta(\tau))$ as the objective function, where $\tau$ denotes a trajectory. For the case of PPO-style online RL algorithms, generally, the policy denominator uses the rollout policy $\pi_{\theta_{old}}(\tau)$.

**Advantage Estimate** $\hat{A}$    In traditional RL, the advantage evaluates the additional benefit of taking the current action given the state. Similarly, the post-training process seeks to maximize the likelihood of generating positive sequences with high advantage and minimize negative sequences.

**Likelihood Gradient** $\nabla\pi_\theta(\tau)$    Likelihood gradient is a general term which maps gradient information from the actions to the model parameters $\theta$. It is crucial for back-propagating the objective signals to the network weights, and is kept the same formal structure across all gradient calculations.

Within UPGE, existing post-training algorithms can be analysed under these four components respectively, clarifying their distinctive properties, which is presented in detail in Appendix C.

## 2.2 Derivation of the Unified Policy Gradient Estimator

In this section, we theoretically derive the UPGE from a common objective shared by all post-training algorithms: improve the likelihood of positive trajectories and decrease that of negative trajectories such that the total reward in expectation $\max_\theta \mathcal{J}(\theta) := \mathbb{E}[r(\tau|q)]$ is maximized. We then show that SFT and RL objectives are not in conflict, and they can be optimized jointly within a single loss.

**Common Objective.**    We model the post-training as a process to maximize the expected success rate while keeping the model policy closely adhering to a demonstration dataset (behavior policy) $\pi_\beta$:

$$\mathcal{J}_\mu(\theta) = \mathbb{E}_{\tau\sim\pi_\theta(\cdot|q)}\big[r(\tau \mid q)\big] \; - \; \mu\,\mathrm{KL}\big(\pi_\beta(\cdot \mid q)\,\|\,\pi_\theta(\cdot \mid q)\big), \qquad \mu \geq 0, \tag{1}$$

where $q \sim \mathcal{D}$ denotes the question from a given distribution, $\tau$ denotes a trajectory, $r$ denotes the (binary/real) score, and $\pi_\beta$ denotes behavior policy from demonstration.

**Gradient of Common Objective.** Differentiating Eq.1 (full derivation in Appendix D.1), we obtain

$$\nabla_\theta \mathcal{J}_\mu(\theta) = \mathbb{E}_{\tau \sim \pi_\theta}\Big[ r(\tau \mid q)\, \nabla_\theta \log \pi_\theta(\tau \mid q) \Big] \; + \; \mu\, \mathbb{E}_{\tau \sim \pi_\beta}\big[ \nabla_\theta \log \pi_\theta(\tau \mid q) \big]. \tag{2}$$

**From gradient to UPGE.** Applying measure-change identity (detailed in Appendix D.1) with reference policy $\pi_{ref}$ mentioned in Section 2.1 and using $\nabla \log \pi_\theta = (1/\pi_\theta)\nabla \pi_\theta$ yields the gradient:

$$\nabla_\theta \mathcal{J}_\mu(\theta) = \mathbb{E}_{\tau \sim \pi_{ref}(\cdot\mid q)}\left[ \frac{1}{\pi_{ref}(\tau \mid q)}\, \widehat{A}_{uni}(\tau, q)\, \nabla_\theta \pi_\theta(\tau \mid q) \right], \tag{3}$$

$$\widehat{A}_{uni}(\tau, q) = \underbrace{r(\tau \mid q)}_{\widehat{A}_{\mathrm{RL}}(\tau,q)} + \underbrace{\mu\, \mathbb{1}\{\pi_{ref} = \pi_\beta\}\, \frac{\pi_\beta(\tau \mid q)}{\pi_\theta(\tau \mid q)}}_{\widehat{A}_{\mathrm{SFT}}(\tau,q)}. \tag{4}$$

In many RL works, the raw score $r(\tau \mid q)$ is replaced by a more structured advantage to reduce variance, provide relative credit assignment within a rollout group, and stabilize step sizes. For example, GRPO uses group-wise normalization: $\widehat{A}_{\mathrm{GRPO}}(\tau_j, q) = \frac{R(\tau_j) - \mathrm{mean}(\{R(\tau)\}_{G_{\mathrm{on}}})}{\mathrm{std}(\{R(\tau)\}_{G_{\mathrm{on}}})}$.

When trust-region stabilization masks, as induced by PPO clipping (detailed in Appendix D.3), are inserted multiplicatively without altering the target objective, we obtain our UPGE:

$$\mathrm{grad}_{uni} = \mathbb{E}_{\tau \sim \pi_{ref}(\cdot\mid q)}\left[ \mathbb{1}_{stable}(\tau, q)\, \frac{1}{\pi_{ref}(\tau \mid q)}\, \widehat{A}_{uni}(\tau, q)\, \nabla_\theta \pi_\theta(\tau \mid q) \right] \tag{5}$$

Details of the derivation above are provided in Appendix D. The gradient in Eq. 2 is the sum of a reward term sampled from $\pi_\theta$ and a data-adherence (SFT) term sampled from $\pi_\beta$. Both terms map to the same estimator via Eq. 3–5 by choosing $\pi_{ref}$ accordingly. (e.g., $\pi_{\theta_{old}}$ for on-policy trust-region updates and $\pi_\beta$ for SFT/offline updates). Therefore, SFT and RL optimize a single Common Objective (Eq. 1) and can be trained jointly within one loss without intrinsic conflict. While these algorithms share the Common Objective, they retain distinct characteristics, and bias-variance trade-offs still exist for different components of the unified gradient estimator, as shown in Appendix C. This observation suggests that an ideal post-training method should adaptively select among methods under varying conditions.

## 2.3 HYBRID POST-TRAINING WITH PERFORMANCE FEEDBACK

Our unified perspective above shows that different post-training losses have the same optimization objective with different characteristics. Inspired by this view, we propose the Hybrid Post-Training (HPT) algorithm. We use a mixed loss $\mathcal{L} = \alpha \mathcal{L}_{\mathrm{RL}} + \beta \mathcal{L}_{\mathrm{SFT}}$, which contains the weighted on-policy RL loss $\mathcal{L}_{\mathrm{RL}}$ and SFT loss $\mathcal{L}_{\mathrm{SFT}}$, to optimize the target LLM $\pi_\theta$. The weights of the two losses ($\alpha$ and $\beta$) are determined by the real-time sampling performance of the model.

**Performance on Single Question.** For each question $q$ provided to LLM, we first obtain supervising trajectory $\tau^\star$ and model's performance $P$ on $q$. Specifically, we draw $n$ on-policy trajectories $\{\tau_i\}_{i=1}^n \sim \pi_\theta(\cdot \mid q)$ and evaluate them with a verifier $v : \tau_i \to \{0, 1\}$, which is the same as the rule-based reward function. Performance $P$ is defined as the mean of these $n$ verification scores:

$$P = \frac{1}{n}\sum_{i=1}^n v(\tau_i), \quad v(\tau_i) = R(\tau_i) = \begin{cases} 1 & \text{if } \tau_i \text{ contains the correct answer of } q \\ 0 & \text{otherwise} \end{cases} \tag{6}$$

**Feedback Coefficients.** Then, we obtain the coefficients based on the performance feedback:

$$\alpha = f(P), \quad \beta = g(P), \tag{7}$$

where the $f$ and $g$ are the specific feedback functions. Experientially, when the model demonstrates strong capability, it is advantageous to emphasize on-policy RL to foster exploration; conversely, when the model's competence is limited, SFT should take precedence to ensure correct guidance. Consequently, $f$ ought to increase with $P$, while $g$ should decrease. In this paper, we employ a pair of simple yet empirically effective switch functions $f$ and $g$:

$$\alpha = f(P) = \begin{cases} 1 & \text{if } P > \gamma \\ 0 & \text{if } P \le \gamma \end{cases}, \quad \beta = g(P) = \begin{cases} 1 & \text{if } P \le \gamma \\ 0 & \text{if } P > \gamma \end{cases} \tag{8}$$

The switch gate $\gamma$ enables the model to perform SFT when its performance falls below a predefined threshold, and RL otherwise.

**Mixed Loss.** Finally, we calculate the RL loss $\mathcal{L}_{\mathrm{RL}}$ using Dr. GRPO with the already generated $n$ on-policy trajectories $\tau_i$ and SFT loss $\mathcal{L}_{\mathrm{SFT}}$ with the supervising trajectory $\tau^\star$:

$$\mathcal{L}_{\mathrm{RL}} = -\frac{1}{n} \sum_{i=1}^{n} \sum_{t=1}^{|\tau_i|} \min\left( r_{i,t}\, A_{i,t},\ \mathrm{clip}\left( r_{i,t},\, 1-\epsilon,\, 1+\epsilon \right) A_{i,t} \right) \tag{9}$$

$$\mathcal{L}_{\mathrm{SFT}} = -\frac{1}{|\tau^\star|} \sum_{t=1}^{|\tau^\star|} \log \pi_\theta(\tau_t^\star \mid q, \tau_{<t}^\star) \tag{10}$$

where $r_{i,t} = \frac{\pi_\theta(\tau_{i,t}|q,\tau_{i,<t})}{\pi_{\theta_{old}}(\tau_{i,t}|q,\tau_{i,<t})}$ is the per-token importance sampling ratio, $A_{i,t} \equiv A_i = \frac{R(\tau_i) - \mathrm{mean}(\{\ R(\tau_i)\ |\ i=1,2,...,n\})}{\mathrm{std}(\{\ R(\tau_i)\ |\ i=1,2,...,n\})}$ is advantage and $\epsilon$ is the clip gate. The mixed loss is then obtained by taking a weighted average of these two losses using performance feedback coefficients $\alpha$ and $\beta$:

$$\mathcal{L} = \alpha \mathcal{L}_{\mathrm{RL}} + \beta \mathcal{L}_{\mathrm{SFT}} \tag{11}$$

The algorithm procedure and detailed analysis of the effectiveness and advantages of our HPT are provided in Appendix E.

## 3 MAIN EXPERIMENTS

### 3.1 EXPERIMENTAL SETUP

**Models**  To evaluate the generalizability of HPT across different models, we conduct experiments using Qwen and LLaMA models of various scales. The models we experiment with are Qwen2.5-Math-1.5B, Qwen2.5-Math-7B (Yang et al., 2024), and LLaMA-3.1-8B (Grattafiori et al., 2024).

**Evaluation Setup**  We evaluate HPT on 6 mathematical reasoning benchmarks: AIME 2024 (Li et al., 2024), AIME 2025 (Li et al., 2024), AMC (Li et al., 2024), MATH-500 (Hendrycks et al., 2021a), Minerva (Lewkowycz et al., 2022), and OlympiadBench (He et al., 2024). Moreover, when employing Qwen2.5-Math-7B as the backbone, we further conduct evaluations on ARC-c (Clark et al., 2018) and GPQA-Diamond (Rein et al., 2024). For the main experiments, following DeepSeek-R1 (Guo et al., 2025), we adopt the *Pass@k* evaluation protocol (Chen et al., 2021) and report *Pass@1* using non-zero temperature sampling. To ensure a fair comparison with previous works (Yan et al., 2025; Fu et al., 2025), we compute avg@32 for AIME 24, AIME 25, and AMC (avg@1 for others) using a temperature of $0.6$ and a top-$p$ value of $0.95$ for accuracy calculation.

**Baselines**  Since HPT dynamically integrates GRPO (Shao et al., 2024) and SFT, the most natural baselines are SFT and GRPO individually. Furthermore, we compare HPT against the mix-policy approach LUFFY (Yan et al., 2025). For experiments using Qwen2.5-Math-7B as the backbone, we additionally include SFT→GRPO and SRFT[1] (Fu et al., 2025) as a baseline, as well as models trained with the Zero-RL procedure on the same backbone for a more comprehensive comparison. We also use PRIME-Zero (Cui et al., 2025a), SimpleRL-Zero (Zeng et al., 2025c), OpenReasoner-Zero (Hu et al., 2025b) and Oat-Zero (Liu et al., 2025b) as baselines.

**Implementation Details**  We fix the switch gate $\gamma$ at 0 throughout all experiments on the Qwen Family models and 2 for LLaMA, and provide relative ablation studies in Section G.2. For hyperparameters, we use a constant learning rate of $5 \times 10^{-6}$ and adopt the AdamW optimizer for the policy model. For rollout, we sample 8 responses using a temperature of $1.0$. The maximum generation length is set to $8,192$ tokens for all models. Other details are reported in Appendix F.

### 3.2 MAIN RESULTS

Table 2 presents the overall performance of HPT on Qwen2.5-Math-7B. HPT not only significantly outperforms both SFT-only and GRPO-only baselines, but also surpasses SFT→GRPO, which requires substantially higher computational cost. This suggests that simply concatenating the two

---

[1]The results of SRFT are based on our own implementation, as the official code is not public.

Table 2: In-distribution and out-of-distribution performance of our HPT and baselines on the Qwen2.5-Math-7B. * means the results are taken from the corresponding paper.

| Model | In-Distribution | | | | | | | Out-of-Distribution | | |
|---|---|---|---|---|---|---|---|---|---|---|
| | AIME 24 | AIME 25 | AMC | MATH-500 | Minerva | Olympiad | Avg | ARC-c | GPQA | Avg |
| Qwen2.5-Math-7B | 12.3 | 4.7 | 33.0 | 43.6 | 8.8 | 13.6 | 19.3 | 30.9 | 28.3 | 29.6 |
| SFT | 25.1 | **22.8** | 56.1 | 84.2 | 33.8 | 44.7 | 44.5 | 67.4 | 25.3 | 46.4 |
| GRPO | 19.4 | 13.8 | 59.1 | 81.8 | 38.2 | 46.2 | 43.1 | 81.2 | 36.4 | 58.8 |
| SFT → GRPO | 25.7 | 21.6 | 62.2 | 84.6 | 38.2 | 46.8 | 46.5 | 67.7 | 30.8 | 49.3 |
| LUFFY | 26.1 | 21.8 | 66.2 | 88.4 | 41.9 | 54.1 | 49.8 | 80.8 | 39.4 | 60.1 |
| SRFT | 18.4 | 15.5 | 55.9 | 83.8 | 42.6 | 48.9 | 44.2 | 80.5 | 36.8 | 58.7 |
| HPT | **33.0** | 21.9 | **69.4** | **89.2** | **46.0** | **56.9** | **52.7** | 81.6 | 42.9 | 62.3 |
| Qwen2.5-Math-7B-Ins. | 11.8 | 9.8 | 48.3 | 83.2 | 34.2 | 39.3 | 37.8 | 72.7 | 29.3 | 51.0 |
| PRIME-Zero* | 17.0 | 12.8 | 54.0 | 81.4 | 39.0 | 40.3 | 40.8 | 73.3 | 18.2 | 45.8 |
| SimpleRL-Zero* | 27.0 | 6.8 | 54.9 | 76.0 | 25.0 | 34.7 | 37.4 | 30.2 | 23.2 | 26.7 |
| OpenReasoner-Zero* | 16.5 | 15.0 | 52.1 | 82.4 | 33.1 | 47.1 | 41.0 | 66.2 | 29.8 | 48.0 |
| Oat-Zero* | 33.4 | 11.9 | 61.2 | 78.0 | 34.6 | 43.4 | 43.8 | 70.1 | 23.7 | 46.9 |

training stages is not the most effective strategy. Moreover, HPT achieves marked improvements over existing mixed-policy approaches such as LUFFY and SRFT, with particularly notable gains of 6.9 and 14.6 points on AIME 2024, respectively. Furthermore, we conduct experiments on models of different scales and families to evaluate the effectiveness of HPT, including LLaMA3.1-8B and Qwen2.5-Math-1.5B, as shown in Table 3. Compared with SFT, GRPO, and LUFFY, HPT achieves substantial performance gains, demonstrating robustness across models with varying capability levels.

Table 3: Performance of HPT and baselines on LaMA3.1-8B and Qwen2.5-Math-1.5B. * means the results are taken from the LUFFY paper (Yan et al., 2025).

| Model | AIME 24 | AIME 25 | AMC | MATH-500 | Minerva | Olympiad | Avg |
|---|---|---|---|---|---|---|---|
| LLaMA3.1-8B | 0.4 | 0.1 | 4.7 | 13.8 | 4.8 | 3.9 | 4.6 |
| SFT* | 0.5 | 0.1 | 5.4 | 20.2 | 4.0 | 5.3 | 5.9 |
| GRPO* | 0.3 | 0.5 | 9.4 | 23.4 | 17.6 | 6.1 | 9.6 |
| LUFFY* | 1.9 | 0.1 | 13.5 | 39.0 | 15.1 | 9.6 | 13.2 |
| HPT | **2.1** | **1.2** | **18.6** | **47.8** | **18.8** | **20.4** | **18.2** |
| Qwen2.5-Math-1.5B | 2.8 | 6.1 | 24.5 | 32.8 | 11.0 | 16.4 | 15.6 |
| SFT | 14.7 | 17.6 | 45.4 | 78.4 | 29.4 | 35.7 | 36.9 |
| GRPO | 12.2 | 8.5 | 43.8 | 71.0 | 33.1 | 35.3 | 34.0 |
| LUFFY | 14.1 | 9.4 | 43.5 | 75.2 | 26.1 | 39.7 | 34.7 |
| HPT | **16.6** | **17.8** | **51.0** | **81.0** | **37.5** | **47.3** | **41.9** |

## 4 EMPIRICAL ANALYSIS

Our empirical analysis progressively reveals how HPT reconciles exploration and exploitation, stabilizes training, and ultimately enhances the reasoning ability. We begin in § 4.1 with an examination of *exploration* and *exploitation*. In § 4.2, we provide a training visualization, contrasting HPT with the conventional SFT→GRPO. Finally, § 4.3 investigates fine-grained training metrics of HPT.

### 4.1 EXPLORATION AND EXPLOITATION

HPT inherently achieves an adaptive switching between RL and SFT. These two paradigms naturally correspond to the learning modes of *exploration* and *exploitation*. Accordingly, we can examine whether HPT addresses the initial challenges from both perspectives.

**Exploration**   From the exploration perspective, we want to analyze the model's *Pass@k* performance after training with HPT. Recently, Limit-of-RLVR (Yue et al., 2025) demonstrated that while RLVR yields an improvement in *Pass@1*, it does not lead to gains in large-$k$ *Pass@k*. In other words, RLVR does not expand the exploratory capability boundary of the base model. We follow Yue et al. (2025) to evaluate *Pass@k* up to 1024 for each question of AIME25, AIME24, and AMC for *Pass@k* evaluation. Based on these sets of generated solutions, we apply bootstrap sampling to obtain accurate estimates of *Pass@k* scores for various values of $k$. Figure 2 illustrates the resulting *Pass@k* curves.

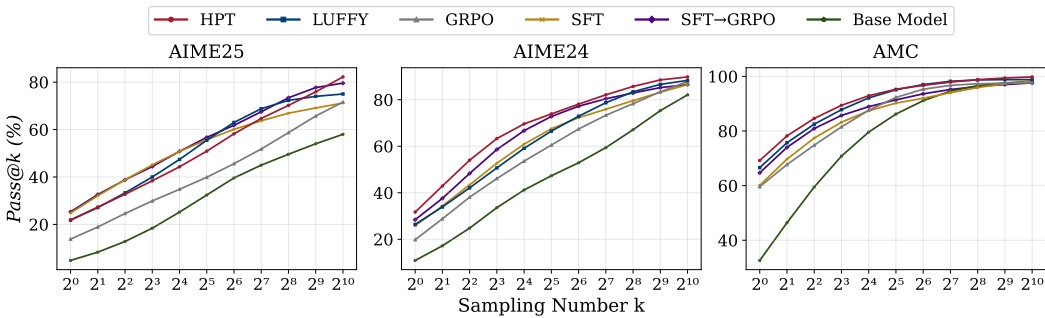

Figure 2: *Pass@k* performance of HPT against baselines on Qwen2.5-Math-7B. Evaluation spans 3 benchmarks, with *Pass@k* values estimated via bootstrap sampling from a set of 2048 generated solutions per question.

- First, we can observe that methods incorporating SFT achieve higher large-$k$ *Pass@k* compared to the GRPO (purely RL). This may be attributed to the introduction of data outside the model's own distribution during SFT, which increases output uncertainty while also providing new knowledge from offline data, thereby enhancing the model's exploratory capacity.

- Furthermore, we identify an interesting phenomenon: since HPT dynamically integrates RL (GRPO) with SFT, we might intuitively expect its large-$k$ *Pass@k* performance to fall between that of the two individual methods. However, HPT achieves the highest large-$k$ *Pass@k* performance overall. This indicates that *Hybrid Post-Training not only delivers substantial improvements in Pass@1, but also maximally preserves and enhances the model's exploratory ability.*

Table 4: Bidirectional analysis of exclusive solves on MATH-500 with Qwen2.5-Math-7B as the backbone model, comparing the HPT against baselines (GRPO and LUFFY). The notation +X / -Y in each cell indicates the performance trade-off: +X represents the number of problems solved by the HPT but not the baseline, while -Y represents the number solved by the baseline but not by the HPT.

| Methods | Level 1 (N=43) | Level 2 (N=90) | Level 3 (N=105) | Level 4 (N=128) | Level 5 (N=134) | Overall (N=500) |
|---|---|---|---|---|---|---|
| **GRPO** | | | | | | |
| Absolute | +0/-0 | +5/-1 | +9/-2 | +17/-4 | +27/-8 | +58/-15 |
| Percentage | +0.0%/-0.0% | +5.6%/-1.1% | +8.6%/-1.9% | +13.3%/-3.1% | +20.1%/-6.0% | +11.6%/-3.0% |
| **LUFFY** | | | | | | |
| Absolute | +1/-0 | +5/-1 | +5/-3 | +10/-5 | +22/-7 | +43/-16 |
| Percentage | +2.3%/-0.0% | +5.6%/-1.1% | +4.8%/-2.9% | +7.8%/-3.9% | +16.4%/-5.2% | +8.6%/-3.2% |

**Exploitation** From the exploitation perspective, the key question is whether HPT, by leveraging SFT, enhances the model's initial competence and facilitates subsequent RL training. To investigate this, we analyze its exclusive solves against the GRPO and LUFFY, building upon the results from the evaluation on MATH-500, as shown in Table 4. The red numbers denote problems that are solved by our HPT but not by GRPO or LUFFY, i.e., problems newly acquired through our training procedure.

- First, the red counts consistently increase with problem difficulty, suggesting that HPT improves the model's ability to tackle more challenging problems. And the red counts are consistently large relative to both baselines, demonstrating that our method enables the model to acquire a substantial number of problems that prior approaches struggled to solve.

- Furthermore, the green counts remain essentially unchanged across settings, which indicates that, compared with existing methods, HPT preserves performance on problems that the model could already solve, thereby mitigating the risk of catastrophic forgetting.

## 4.2 TRAINING VISUALIZATION

To facilitate a fine-grained examination of the training process and obtain deeper insights into how HPT works, we conduct a visualization analysis comparing the SFT→GRPO with HPT. We sample 255 questions from the MATH dataset (Hendrycks et al., 2021b) for training, with 85 each from Levels 3, 4, and 5. For SFT→GRPO, we perform 50 epochs of GRPO on a Qwen2.5-Math-1.5B

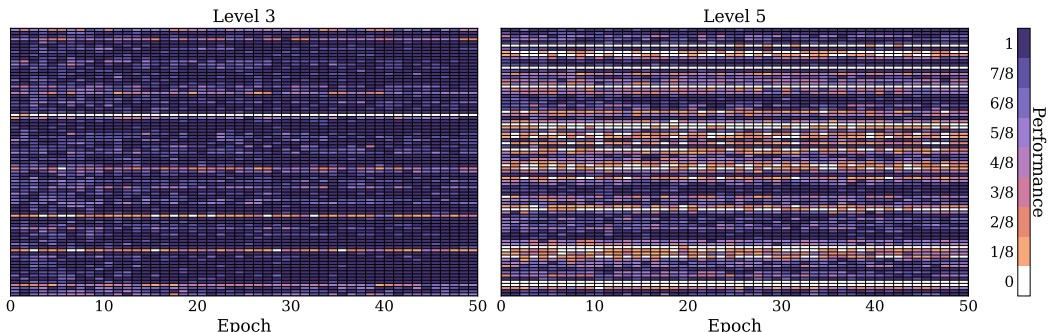

Figure 3: GRPO training dynamics of SFT→GRPO on Qwen2.5-Math-1.5B across 50 epochs. Each line represents a question and we visualize the model's per-question rollout accuracy throughout the training process.

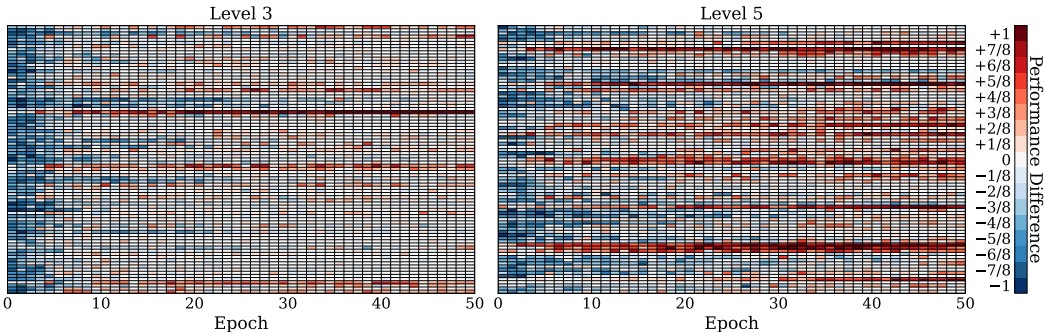

Figure 4: Performance difference (HPT v.s. SFT→GRPO) on Qwen2.5-Math-1.5B across 50 epochs. Each line represents a question. A diverging color scale indicates the advantage: red for HPT, blue for SFT→GRPO.

model fine-tuned with SFT, tracking rollout accuracy of each question across training, as shown in Figure 3. To highlight difficulty effects, we focus on Levels 3 and 5 as representative cases. Notably, GRPO frequently produces continuous white lines, reflecting widespread rollout errors across outputs. This illustrates a core limitation of RL methods: they struggle to learn effectively when frequent rollout errors occur.

In parallel, we use HPT to train Qwen2.5-Math-1.5B from scratch. To enable a more intuitive comparison, we calculate the accuracy difference at matched questions and epochs in the evaluation grid between two methods: red indicates HPT is better, blue the opposite. Figure 4 presents the results of the difference plots. Notably, SFT→GRPO, which involves a preceding SFT phase, requires greater computational resources than HPT. This additional computation leads to an initial dominance of the blue regions, considering the SFT stage in SFT→GRPO has already incorporated substantial prior knowledge. However, in the later training stage, HPT still surpasses and ultimately reveals the dominance of the red regions, indicating its superiority. This advantage becomes even more pronounced in the Level 5 subplot, suggesting that HPT provides particular benefits for learning on more challenging problems, which may be due to its appropriate use of the demonstration data.

## 4.3 TRAINING DYNAMICS

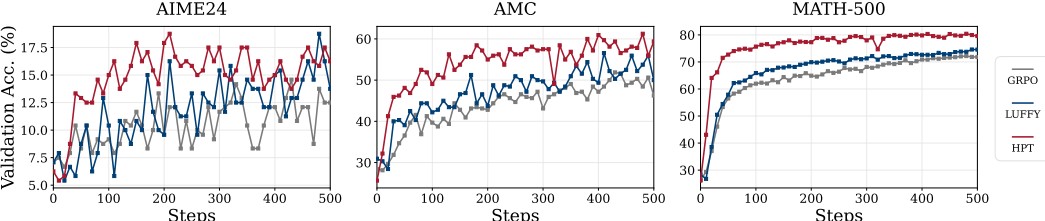

Figure 5: Validation performance comparisons on Qwen2.5-Math-1.5B across AIME24, AMC and MATH-500.

**Validation Performance.** We track the validation performance on Qwen2.5-Math-1.5B as shown in Figure 5, where HPT consistently outperforms the baselines and delivers stable improvements.

**Offline Data Ratio.** As shown in Figure 6, we quantify the offline data ratio, which is defined as the proportion of offline samples relative to the total number of training samples, at each training step. As expected, when the model has not yet acquired competence on the target tasks, the early phase is characterized by a large proportion of SFT-driven updates. As training progresses and the model's on-policy reward increases, the mixture gradually shifts: the contribution of RL grows while that of SFT diminishes, eventually stabilizing at a small but non-zero level. This trend is observed for both Qwen2.5-Math-7B and Qwen2.5-Math-

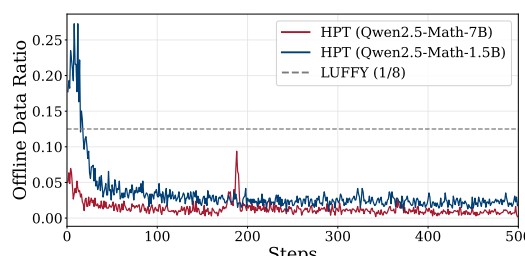

Figure 6: Offline data ratio dynamics during training, which is calculated as the proportion of offline training samples relative to the total training data at each step.

1.5B. The weaker 1.5B model remains in the SFT-dominated regime for a longer period before transitioning, whereas the stronger 7B model shifts earlier. These results align with our analysis of the design of HPT, where the mixing ratio is automatically adjusted based on performance rather than fixed in advance like LUFFY.

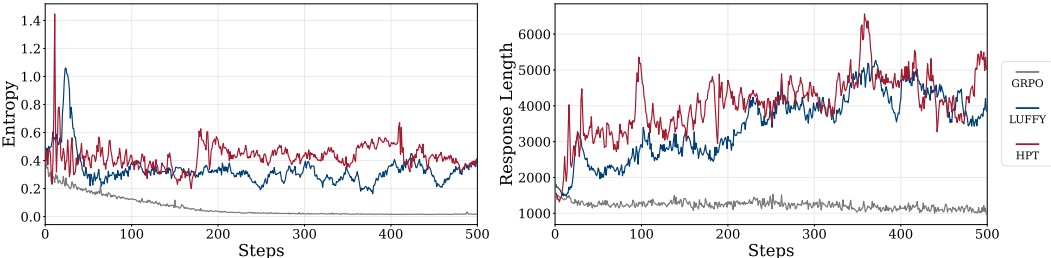

Figure 7: Comparisons of training dynamics: (left) The entropy measures the diversity of model outputs, indicating exploration behavior. (right) The response length tracks the average length of generated responses.

**Entropy and exploration.** Figure 7 (left) tracks token-level entropy over 500 steps. HPT maintains higher entropy than GRPO throughout the training phases. This is expected as the offline SFT trajectories are derived from the external demonstration distribution, which consequently increases the diversity in the model's outputs.

**Response length and acquired reasoning patterns.** Figure 7 (right) reports the average response length. Our SFT trajectories have a length of up to 8k tokens. Under HPT, the model's response length increases quickly during the early steps but does not jump to the 8k ceiling. More importantly, after the method shifts toward RL and the SFT proportion plateaus at a low level, the response length does not regress. This persistence suggests that the model has internalized long-form reasoning routines from the offline data rather than merely echoing teacher outputs. In other words, the learned reasoning pattern becomes part of the policy, and RL fine-tuning refines it instead of erasing it.

## 5 CONCLUSION

In this paper, we introduce the Unified Policy Gradient Estimator to unify the post-training of LLMs. We demonstrate that SFT and RL optimize a common objective, with their gradients representing different bias-variance tradeoffs. Motivated by this unified perspective, we propose Hybrid Post-Training (HPT), an algorithm that dynamically adapts between SFT for exploitation and RL for exploration based on real-time performance feedback. Extensive empirical validation shows that HPT consistently outperforms strong baselines across various models and benchmarks. Our work contributes both a unifying theoretical perspective on post-training and a practical algorithm that effectively balances exploitation and exploration to enhance model capabilities.

ETHICS STATEMENT

This paper introduces UPGE to unify the post-training of LLMs and the mix-policy algorithm HPT. We study post-training for LLM reasoning on public math/logic benchmarks only, and no human subjects or personal data are used. We think our work doesn't introduce any ethical concerns.

REPRODUCIBILITY STATEMENT

We provide the implementation code and scripts of our HPT algorithm and main baselines in the Supplementary Material for reproduction. And we also specify the experimental configurations and hyperparameters in detail in Section 3 and Appendix F.

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

## A    MORE RELATED WORKS

### A.1    LLM POST-TRAINING: SFT AND RL

Current post-training methodologies for LLMs are largely centered around two primary paradigms: Supervised Fine-Tuning (SFT) and Reinforcement Learning (RL) (Wei et al., 2021; Ouyang et al., 2022). In the SFT paradigm, models are adapted for specific applications through training on curated input-output pairs, a process which has been shown to effectively align their behavior with human demonstrations (Chung et al., 2022; Longpre et al., 2023; Touvron et al., 2023a;b). In parallel, numerous works have highlighted RL as an effective approach for refining LLM behavior in ways that are difficult to capture with SFT's static datasets (Glaese et al., 2022; Bai et al., 2022; Nakano et al., 2021). Within this domain, a popular framework is Reinforcement Learning from Human Feedback (RLHF), which optimizes the LLM policy against a reward model trained on human preferences (Christiano et al., 2017; Stiennon et al., 2020). Multiple works have established Proximal Policy Optimization (PPO) as a cornerstone algorithm for this phase (Schulman et al., 2017; Ziegler et al., 2019). To further improve reasoning capabilities in reward-driven optimization, recent advancements like Group Relative Policy Optimization (GRPO) have also been developed and widely adopted (Shao et al., 2024; Zheng et al., 2025; Chen et al., 2025).

### A.2    A COMBINATION OF ONLINE AND OFFLINE DATA IN LLM POST-TRAINING

Beyond applying SFT or RL in isolation, further explorations have sought to synergize their respective strengths by combining signals from pre-existing *offline data* and dynamically generated *online data* (Fu et al., 2025; Yan et al., 2025; Ma et al., 2025). This motivation stems from the distinct characteristics of each approach: SFT is noted for its efficiency in distilling knowledge from offline sources, whereas RL is valued for fostering exploration through online rollouts, a process frequently linked to improved generalization (Rajani et al., 2025; Chu et al., 2025). The strategies for this integration are diverse; some techniques use offline data as a prefix to guide online generation (Zhou et al., 2023; Touvron et al., 2024; Li et al., 2025; Wang et al., 2025), while others enhance offline data by incorporating reward signals in a process known as reward-augmented fine-tuning (Liu et al., 2024; Zhao et al., 2023; Park et al., 2025; Sun et al., 2025). The broader landscape also includes various purely offline preference optimization methods, though they follow a different paradigm (Rafailov et al., 2023; Mitchell et al., 2024; Liu et al., 2025c; Ethayarajh et al., 2024; Ahmadian et al., 2024; Wu et al., 2025). However, the most direct approach to synergy involves the concurrent use of both data types for training updates.

This direct approach, often termed mix-policy learning, is particularly relevant to our work and typically involves updating the model with a composite objective that combines an SFT loss from offline data and an RL loss from online data (Dong et al., 2023; Gulcehre et al., 2023; Singh et al., 2023; Liu et al., 2023). For instance, LUFFY (Yan et al., 2025) explores this paradigm by combining a fixed ratio of offline demonstration data with online rollouts in each training batch. Subsequently, SRFT (Fu et al., 2025) proposed a monolithic training phase that dynamically adjusts the weights of SFT and RL losses based on the model's policy entropy, further demonstrating the viability of unifying these signals over a sequential pipeline. The principle of creating such a composite loss is shared by a variety of other recent frameworks (Wu et al., 2024; Zhang et al., 2025; Kim et al., 2025; Yu et al., 2025; Liu et al., 2025a; He et al., 2025a). While these methods highlight a clear trend towards unifying training signals, a foundational theoretical analysis explaining why these different learning signals can be effectively combined is still lacking. This motivates our work to establish a unified theoretical framework that in turn inspires a more principled algorithm design.

## B    GRADIENT DERIVATION FOR CLASSICAL ALGORITHMS

### B.1    GRADIENT OF SFT

We first consider the SFT process as a warm-up. As mentioned in the previous section, SFT takes a pre-trained foundation model and further makes the model more specialized by training its output prediction distribution to align with domain-specific data. The fine-tuning process uses the same cross-entropy loss as in model pre-training, defined as follows,

$$\mathcal{L}_{SFT}(\theta) = -\sum_{i=1}^{N}\sum_{t=1}^{|\tau_i|} \log \pi_\theta(\tau_{i,t}|q_i, \tau_{i,<t}). \tag{12}$$

where $\mathcal{D}_{SFT} = \{(q_i, \tau_i)\}_{i\in[N]}$ denotes the SFT dataset consisting of $N$ question and trajectory pairs. $\tau_t$ denotes the $t$-th token in the trajectory and $\tau_{<t}$ denotes all the tokens prior to $\tau_t$.

For any $t$, the LLM outputs the next-token prediction as a probability distribution. In the context of RL, such a probability distribution has been commonly considered as a stochastic policy. Then, the gradient calculation of SFT can be obtained by directly taking the derivative of Equation (12) and takes the following form:

$$\nabla \mathcal{J}_{SFT}(\theta) = -\nabla \mathcal{L}_{SFT}(\theta) = \sum_{i=1}^{N}\sum_{t=1}^{|\tau_i|} \nabla\pi_\theta(\tau_{i,t}|q_i, \tau_{i,<t}) \frac{1}{\pi_\theta(\tau_{i,t}|q_i, \tau_{i,<t})}. \tag{13}$$

In this section, we slightly abuse the notion of policy gradient and consider the SFT as a case of behavioral cloning (BC) (Torabi et al., 2018), and Equation (13) can be seen as a specific form of policy gradient.

## B.2 GRADIENT OF ONLINE RL: PPO, GRPO AND BEYOND

For online RL, we first consider Proximal Policy Optimization (PPO) (Schulman et al., 2017) and a series of its derivations. PPO is a pivotal technique for RLVR in LLMs. Motivated by TRPO, PPO keeps the new policy close to the old policy, and performs conservative policy updates by incorporating a clipped version of its policy ratio in its objective. The clipping function was shown to stabilize the training process and avoid performance collapse during training. In this section, we omit the regularization terms, such as the KL divergence and entropy. The loss objective for PPO can be written as follows,

$$\mathcal{L}_{PPO}(\pi_\theta) = -\frac{1}{N}\sum_{i=1}^{N}\frac{1}{G}\sum_{j=1}^{G}\frac{1}{|\tau_j|}\sum_{t=1}^{|\tau_j|} \min(r_{i,j,t}(\theta)\hat{A}_{i,j}, \text{clip}(r_{i,j,t}(\theta), 1-\epsilon, 1+\epsilon)\hat{A}_{i,j}), \tag{14}$$

In this setting, we consider questions sampled from a given dataset $\mathcal{D}_{RL} \triangleq \{q_i\}_{i=1}^{N}$, and for each question, we consider $G$ trajectories independently sampled using a reference policy $\pi_{ref}$. We use $r_{i,j,t}(\theta) = \frac{\pi_\theta(\tau_{i,j,t}|q_i,\tau_{i,j,<t})}{\pi_{ref}(\tau_{i,j,t}|q_i,\tau_{i,j,<t})}$ to denote the policy ratio $\pi_\theta/\pi_{ref}$ introduced for importance sampling, $\epsilon$ denotes the clipping factor for the importance sampling ratio, enhancing stability.

For PPO, $\hat{A}$ is estimated using the Generalized Advantage Estimation (GAE) (Schulman et al., 2015b), calculated based on the reward of the sampled trajectories. For the case of GRPO, the advantage estimate $\hat{A}$ is calculated based on a set of sampled trajectories. Given question $q_i$, a group of sampled roll-out trajectoried $\{\tau_{i,j}\}_{j\in[G]}$ with verifiable reward $R(\tau_{i,j}) \in \{0,1\}$, $\hat{A}_{i,j}$ is calculated as the normalized reward over the group.

$$\hat{A}_{i,j} = \frac{R(\tau_{i,j}) - \text{mean}(\{R(\tau_{i,k})\}_{k\in[G]})}{\text{std}(\{R(\tau_{i,k})\}_{k\in[G]})}, \tag{15}$$

Compared to PPO, the most significant difference introduced by GRPO is the group relative advantage described above. Notably, the original manuscript of GRPO has also induced a sequence-level policy gradient balancing and a KL regularization term. However, more recent works such as (Yu et al., 2025) have removed or modified these terms in general.

The clipped surrogate objective in PPO and similar algorithms enhances the stability of the RL training process by turning off gradient propagation on samples where $\pi_\theta$ moves too far from $\pi_{ref}$. For gradient calculation, this can be represented as an indicator function $\mathbf{1}_{clip}$.

$$\nabla \mathcal{J}_{PPO} = -\nabla \mathcal{L}_{PPO} = \frac{1}{N}\sum_{i=1}^{N}\frac{1}{G}\sum_{j=1}^{G}\frac{1}{|\tau_j|}\sum_{t=1}^{|\tau_j|} \nabla\pi_\theta(\tau_{i,j,t}|q_i, \tau_{i,j,<t})\frac{\hat{A}_{i,j}\mathbf{1}_{clip}}{\pi_{ref}(\tau_{i,j,t}|q_i, \tau_{i,j,<t})}. \tag{16}$$

Apart from PPO and GRPO, many recent RL algorithms for RL post-training in LLMs can be shown to exhibit a similar form for their policy gradient calculations.

### B.3 GRADIENT OF OFFLINE RL

As stated in the previous sections, many recent studies seek to leverage offline data in the online RL training process for LLMs. These methods consider expert demonstration data as trajectories sampled from a near-optimal policy, and perform RL updates on these data based on policy gradient updates. These algorithms are adapted from the online RL literature and often combine offline and online training, setting them apart from simple SFT.

Taking SRFT (Fu et al., 2025) as an instance, the offline RL objective can be written as follows

$$\mathcal{L}_{SRFT}(\pi_\theta) = -\frac{1}{N}\sum_{i=1}^{N}\frac{1}{G}\sum_{j=1}^{G}\frac{1}{|\tau_j|}\sum_{t=1}^{|\tau_j|}\pi_\theta(\tau_{i,j,t}|q_i,\tau_{i,j,<t})\hat{A}_{i,j},$$

(17)

This objective is derived from the GRPO objective in Equation (14), while setting $\pi_{ref} \equiv 1$ and removing the clipping mechanism since it becomes imbalanced. The motivation behind setting $\pi_{ref} \equiv 1$ is that $\pi_{ref}$ is typically unavailable for offline data. Under the assumption that the demonstration policy evenly covers the current policy $\pi_\theta$. In this case, setting $\pi_{ref}$ to 1 changes the algorithm from importance sampling to rejection sampling. The policy gradient of the offline SRFT objective can be derived consequently.

$$\nabla\mathcal{J}_{SRFT} = -\nabla\mathcal{L}_{SRFT} = \frac{1}{N}\sum_{i=1}^{N}\frac{1}{G}\sum_{j=1}^{G}\frac{1}{|\tau_j|}\sum_{t=1}^{|\tau_j|}\nabla\pi_\theta(\tau_{i,j,t}|q_i,\tau_{i,j,<t})\frac{\hat{A}_{i,j}}{\pi_{ref}=1}.$$

(18)

## C GRADIENT COMPONENT ANALYSIS

> **Takeaways**
>
> 1. While all algorithms share the same Common Objective, bias-variance trade-offs still exist across current instances for different components of the unified gradient estimator.
>
> 2. We can improve the post-training process by constructing a better and more suitable estimation of the policy gradient.

Across the wide spectrum of algorithms contained in our previous discussions and Table 1, it can be inferred that the four components that construct the unified gradient estimator are motivated by different procedures in the post-training process. To better illustrate the relationship between the different processes with the respective components of our unified gradient, we present Figure 1.

We divide the post-training process of LLMs into the four steps shown in Figure 1: i) First, the LLM makes the decision on its data source, either to use data from an offline demonstration dataset, from self-generated rollout data, or a mixture of both. In this process, the policy likelihood $\pi_\theta$ of the data with respect to the current LLM is generated. ii) Given the data source used for data generation, a reference policy $\pi_{ref}$ is calculated. iii) After data collection is complete, the algorithm calculates the advantage estimation $\hat{A}$ for each token/sequence. iv) Lastly, the algorithm may choose to apply an additional masking procedure $\mathbb{1}_{stable}$ to disable the gradient calculation of various tokens, which could lead to theoretical or numerical stability issues. After these four steps, the components are collected to construct the policy gradient $\text{grad}_{Uni}$, which is used to update the LLM in the system. Similar to GAE presented in Schulman et al. (2015b), multiple instantiations exist to estimate the policy gradient. However, different component selections introduce various degrees of bias and variance, where a trade-off is often encountered. We provide the following discussion on key components of the unified gradient below.

**Reference Policy Calculation** Practically speaking, the reference policy denominator places a weight on each token-level update such that any token with a smaller probability, often implying more significance, is weighted more. SFT and REINFORCE assign weights inversely proportional

to the current policy $\pi_\theta$, enforcing a bigger update when the model outputs a small probability. On the other hand, when the data is generated with an outdated model, algorithms such as PPO assign weights inversely proportional to the rollout policy $\pi_{\theta_{old}}$, and offline RL does not assign additional weights for tokens.

Theoretically, the reference policy is usually set given the source of the dataset and/or the rollout policy. For online RL methods that train purely with on-policy data, such as REINFORCE (Ahmadian et al., 2024), uses $\frac{1}{\pi_\theta}$, which produces an unbiased estimate for gradient calculation. However, these methods usually suffer from high variance. For PPO-style online RL algorithms, the reference policy refers to the rollout policy, which is a result of importance sampling. PPO is a numerically simplified version of TRPO (Schulman et al., 2015a). PPO makes conservative updates that effectively reduce variance. However, the important sampling ratio is in fact theoretically ill-posed and could introduce systematic bias, as discussed in GSPO (Zheng et al., 2025). GSPO has also proposed a novel calculation for $\pi_{ref}$, as shown in Table 1. On the other hand, in the offline setting, the choice for reference policy $\pi_{ref}$ is limited, since the algorithm generally has no access to the rollout policy. If we are given the assumption that the offline data evenly covers the entire state-action rollout space, then the importance sampling ratio $r(\theta) = \frac{\pi_\theta(\tau)}{\pi_{ref}(\tau)}$ reduces to $\pi_\theta(\tau)$ by setting constant $\pi_{ref}(\tau) = 1$. Notably, it is apparent that setting $\pi_{ref}(\tau) = 1$ introduces much bias at the cost of numerical stability. For the SFT case, we can consider that the domain-specific dataset is generated with respect to the expert policy $\pi^*$; therefore, no weighted sampling is required. Neither of the two approaches is entirely theoretically justified, from an RL perspective; both require a lower bound on the state-action visitation of all the possible state-action pairs (Kakade, 2003), which can not be satisfied due to the severely limited datasets in practice.

Apart from the strong connection to data source and sampling polices, some studies employ a hand-crafted reweight factor within the reference policy denominator. These works (Yan et al., 2025; Zhang et al., 2025) typically find desirable token properties and purposefully place a higher/lower weight on these desirable/undesirable tokens, respectively.

**Choice of Stabilization Mask**  The clipping operation introduced in PPO was the first to explicitly add a stop gradient operation on LLM post-training. Clipping gradient estimation where the importance sampling strays too far from 1 is an effective approach to address high variances. However, this aggressive clipping behavior has been criticized by some to be overly conservative: Both DAPO (Yu et al., 2025) and CISPO (Chen et al., 2025) stated that the classical PPO approach drops all the tokens corresponding to large model updates, and that many such tokens are in fact crucial for stabilizing entropy and facilitating scalable RL. DAPO presented a slight modification to the clipping threshold, and CISPO further extended the notion of token-wise mask, where more granular tuning was introduced to decide whether gradients from specific tokens should be dropped. The recent work of Cui et al. (2025b) has demonstrated that many existing algorithms negatively impact the output entropy during training and introduced Clip-Cov, adding another clipping mechanism to address the entropy-collapse encountered in training. While these methods demonstrated performance enhancements in practice, they also provide additional sources of bias.

On the other hand, works such as GSPO (Zheng et al., 2025) have stated that the PPO-style clipping is inherently noisy and inefficient for sample exploitation: GSPO clips a much larger fraction of tokens and yet demonstrated superior training efficiency.

In addition, post-training algorithms using offline data have chosen to purposefully remove the clipping from training, mostly guided by performance. Though setting $\pi_{ref}(\tau) = 1$ as the policy denominator does effectively reduce the instability in gradient calculations.

**Advantage Estimation**  There are two commonly used settings for estimating the sequence-level advantage function: the fixed advantage setting and the adaptive advantage setting. The fixed setting considers $\hat{A} = \pm 1$ given the rule-based verification, which is adapted by REINFORCE and implicitly by SFT (where all sequences are positive samples). Alternatively, recent studies have focused on using adaptive advantage estimations, performing re-centering or normalization based on the performance of the current rollout group. Notably, GRPO and its variants, such as DAPO (Yu et al., 2025) and LUFFY (Yan et al., 2025), use unit normalization such that the advantage estimation of the group has a unit standard deviation. Other approaches, such as Dr. GRPO (Liu et al., 2025b), RLOO (Ahmadian et al., 2024), and REINFORCE++ (Hu et al., 2025a), claim that dividing the standard

deviation introduces a difficulty bias and that only recentering is adequate.

Apart from sequence-level advantage estimate $\hat{A}_{i,j}$, recent works (Wang et al., 2025; Yang et al., 2025; Sun et al., 2025) have also adapted a more granular token-level advantage estimate $\hat{A}_{i,j,t}$ to a varying degree of success.

**A Combination of Gradient Estimators**    Although bias-variance trade-offs exist for the gradient estimator, we state that, given data distribution assumptions and sufficient data samples, all policy gradient estimators covered in our framework should result in an effective direction of improvement for the Common Objective. To effectively reduce the variance and bias for each policy update, we can treat instances of policy gradient as different noisy measurements of the true policy gradient, and perform a weighted average to generate a more accurate gradient estimation, similar to complementary filters (Marantos et al., 2015).

However, the complexity of LLM RLVR introduces additional challenges. The current state of the behavior policy $\pi_\theta$ and its relationship with the respective tasks also greatly impacts the bias-variance tradeoff of each instance of the gradient estimator. For instance, RL-zero is significantly more effective for the Qwen model series compared to LlaMA, but SFT is effective for both methods (Zeng et al., 2025b); SFT $\rightarrow$ RL and RL $\rightarrow$ SFT also yield significantly different results on the same LLM (Fu et al., 2025). We argue that for constructing a post-training algorithm with better effectiveness and efficiency, a dynamic and adaptive mechanism is crucial to construct optimal gradient components.

# D    ADDITIONAL THEORETICAL DETAILS FOR SECTION 2.2

## D.1    DERIVING EQUATION 2 FROM EQUATION 1

**Lemma A1** (Score-function identity).    *For density $\pi_\theta$ and integrable $f(\tau)$,*

$$\nabla_\theta \, \mathbb{E}_{\tau\sim\pi_\theta}[f(\tau)] = \mathbb{E}_{\tau\sim\pi_\theta}\big[f(\tau)\,\nabla_\theta \log \pi_\theta(\tau)\big], \qquad \mathbb{E}_{\tau\sim\pi_\theta}\big[\nabla_\theta \log \pi_\theta(\tau)\big] = 0.$$

**Lemma A2** (Differentiating an expectation with parameterized integrand).    *For differentiable $f_\theta$,*

$$\nabla_\theta \, \mathbb{E}_{\tau\sim\pi_\theta}[f_\theta(\tau)] = \mathbb{E}_{\tau\sim\pi_\theta}\big[\nabla_\theta \log \pi_\theta(\tau)\, f_\theta(\tau) + \nabla_\theta f_\theta(\tau)\big].$$

**Lemma A3** (Measure-change (importance reweighting) identity).    *Let $s(\tau \mid q)$ be any sampling density that is positive wherever $\pi_\theta(\tau \mid q)$ is. Then*

$$\mathbb{E}_{\tau\sim\pi_\theta}\big[f(\tau)\,\nabla_\theta \log \pi_\theta(\tau)\big] = \mathbb{E}_{\tau\sim s}\Big[\frac{\pi_\theta(\tau)}{s(\tau)}\, f(\tau)\, \nabla_\theta \log \pi_\theta(\tau)\Big] = \mathbb{E}_{\tau\sim s}\Big[\frac{1}{s(\tau)}\, f(\tau)\, \nabla_\theta \pi_\theta(\tau)\Big].$$

*Proof.* By Lemma A1, $\nabla\mathbb{E}_{\pi_\theta}[r(\cdot \mid q)] = \mathbb{E}_{\pi_\theta}[r(\cdot \mid q)\,\nabla \log \pi_\theta]$. For the data-adherence term, since $\mathrm{KL}(\pi_\beta\|\pi_\theta) = \mathbb{E}_{\pi_\beta}[\log \pi_\beta - \log \pi_\theta]$ and $\pi_\beta$ does not depend on $\theta$, we have $-\mu \, \nabla\mathrm{KL}(\pi_\beta\|\pi_\theta) = \mu \, \mathbb{E}_{\pi_\beta}[\nabla \log \pi_\theta]$. Summing yields the claim.    □

## D.2    EXTENSION: ADDING A TRUST-REGION REGULARIZER

A trust region encourages conservative policy updates by penalizing the KL divergence from the current policy $\pi_\theta$ to a fixed reference policy $\pi_{ref}$:

$$\lambda \, \mathrm{KL}\big(\pi_\theta(\cdot \mid q) \,\|\, \pi_{ref}(\cdot \mid q)\big), \qquad \lambda \geq 0.$$

It is the penalty form of the constrained problem

$$\max_\theta \, \mathbb{E}_{\tau\sim\pi_\theta}[r(\tau \mid q)] \quad \text{s.t.} \quad \mathrm{KL}\big(\pi_\theta\|\pi_{ref}\big) \leq \delta,$$

where $\lambda$ acts as the Lagrange multiplier tied to the trust-region radius $\delta$. Typical choices are $\pi_{ref} = \pi_{\theta_{old}}$ (on-policy stability, TRPO/PPO-style). This penalty controls step sizes, dampens distribution shift, and yields clipping-style masks when optimized with PPO surrogates.

**Objective and gradient with trust region.** Augmenting the Common Objective with the trust-region term gives

$$\widetilde{\mathcal{J}}_{\lambda,\mu}(\theta) = \mathbb{E}_{\tau\sim\pi_\theta(\cdot|q)}[r(\tau \mid q)] - \lambda \, \mathrm{KL}\big(\pi_\theta(\cdot \mid q) \,\|\, \pi_{ref}(\cdot \mid q)\big) - \mu \, \mathrm{KL}\big(\pi_\beta(\cdot \mid q) \,\|\, \pi_\theta(\cdot \mid q)\big),$$

whose gradient is

$$\nabla_\theta \widetilde{\mathcal{J}}_{\lambda,\mu}(\theta) = \mathbb{E}_{\tau \sim \pi_\theta}\Big[\big(r(\tau \mid q) - \lambda \log \tfrac{\pi_\theta(\tau|q)}{\pi_{ref}(\tau|q)}\big)\nabla_\theta \log \pi_\theta(\tau \mid q)\Big] + \mu\,\mathbb{E}_{\tau \sim \pi_\beta}\big[\nabla_\theta \log \pi_\theta(\tau \mid q)\big].$$

In the estimator (Equation 3), this corresponds to replacing the unified advantage by

$$\widehat{A}^{(\lambda)}_{uni}(\tau,q) = r(\tau \mid q)\ -\ \lambda \log \frac{\pi_\theta(\tau \mid q)}{\pi_{ref}(\tau \mid q)}\ +\ \mu\,\mathbb{1}\{\pi_{ref} = \pi_\beta\}\,\frac{\pi_\beta(\tau \mid q)}{\pi_\theta(\tau \mid q)}.$$

All other expressions, including the masked estimator in Equation 5, remain unchanged in form (with $\widehat{A}_{uni}$ replaced by $\widehat{A}^{(\lambda)}_{uni}$).

### D.3 PPO Clipping and the Stabilization Mask

With rollout policy $\pi_{\theta_{old}}$ and trust-region constraint $\mathrm{KL}(\pi_\theta \| \pi_{\theta_{old}}) \leq \delta$, the PPO surrogate

$$\max_\theta\ \ \mathbb{E}_{\tau \sim \pi_{\theta_{old}}}\Big[\min\big(r_\theta(\tau)\,A_{\theta_{old}}(\tau),\ \mathrm{clip}(r_\theta(\tau), 1-\epsilon, 1+\epsilon)\,A_{\theta_{old}}(\tau)\big)\Big],\ \ r_\theta(\tau) = \frac{\pi_\theta(\tau)}{\pi_{\theta_{old}}(\tau)},$$

has a piecewise derivative that is zero outside the trusted region in the harmful direction, yielding

$$\nabla_\theta \approx \mathbb{E}_{\tau \sim \pi_{\theta_{old}}}\left[\mathbb{1}_{stable}(\tau)\,\frac{1}{\pi_{\theta_{old}}(\tau)}\,A_{\theta_{old}}(\tau)\,\nabla_\theta \pi_\theta(\tau)\right], \tag{19}$$

which matches the masked Unified Policy Gradient Estimator with $\pi_{ref} = \pi_{\theta_{old}}$ and $\widehat{A} = A_{\theta_{old}}$.

## E The Procedure and Analysis of Hybrid Post-Training (HPT)

---

**Algorithm 1** The Hybrid Post-Training (HPT) Algorithm

---

**Input:** Pretrained LLM (policy) $\pi_\theta$; SFT dataset $\mathcal{D}_{\mathrm{SFT}} = \{(q, \tau^\star)\}$ with supervising trajectories $\tau^\star$;
      verifier $v$; on-policy samples number $n$; total training steps $T$; feedback functions $f$ and $g$;
      learning rate $\eta$
**Output:** Fine-tuned policy $\pi_{\theta^*}$.
**for** $t = 1$ **to** $T$ **do**
    **for** $i = 1$ **to** $n$ **do**
        Sample trajectory $\tau_i \sim \pi_\theta(\cdot \mid q)$ Evaluate with verifier (rule-based reward):   $v(\tau_i) \leftarrow$
        $R(\tau_i) \in \{0, 1\}$
    **end**
    $P \leftarrow \frac{1}{n}\sum_{i=1}^n v(\tau_i)$  $\alpha \leftarrow f(P),$   $\beta \leftarrow g(P)$ `# Performance feedback on q`
    Compute RL loss $\mathcal{L}_{\mathrm{RL}}$ using rollouts $\{\tau_i\}$ and normalized advantages derived from $\{R(\tau_i)\}$.
    Compute SFT loss $\mathcal{L}_{\mathrm{SFT}}$ on the supervising trajectory $\tau^\star$.
    $\mathcal{L} \leftarrow \alpha\,\mathcal{L}_{\mathrm{RL}} + \beta\,\mathcal{L}_{\mathrm{SFT}}$ `# Mixed loss with feedback coefficients`
    $\theta \leftarrow \theta - \eta\,\nabla_\theta \mathcal{L}$
**end**
**return** $\pi_{\theta^*}$

---

We provide a detailed analysis of the effectiveness and advantages of HPT, the procedure of which is shown above.

**HPT as an Optimal Gradient Selection Mechanism** As stated in Section 2, we can consider the gradient calculations of existing approaches (including SFT, offline RL, and online RL) as different instances of gradient calculation, shown in Table 1. We argue that, similar to the choice between behavior cloning and reinforcement learning in robotics learning (Kumar et al., 2022), the exact gradient calculation should be carefully chosen for the post-training process of reasoning tasks.

When the model's current policy is far away from the optimal policy, SFT, or behavior cloning, should be preferred over RL. Usually, this is indicated by the poor performance of the language model. This is evidently shown in the Reference Policy of SFT in Table 1. The current model policy being far from the demonstration would result in the Reference Policy $\pi_\theta(\tau) << 1$. Therefore, the SFT

gradient could make a significantly larger model update compared to RL methods. Utilizing SFT could speed up the post-training process and increase performance.

On the other hand, when the model exhibits relatively good performance, it implies that the model's current policy is close to the optimal policy, and the online state-action distribution could more effectively cover the optimal response. Theoretically, this can be represented by the concentrability coefficient (Rashidinejad et al., 2021). In this case, online RL is preferred over SFT for its scalability. In addition, (Kumar et al., 2022) has shown that for problems with long-horizon properties and the environment has "critical" states. Both properties are satisfied in our math reasoning task, given recent studies such as Wang et al. (2025).

**Amplification of Success Probability via HPT** By reverting to SFT when the model fails entirely, we ensure policy improvement by injecting the high-quality external signal. When the model already achieves partial success, RL-style updates amplify the probability of success, which is similar to the success amplification property analyzed in RL theory. Therefore, for initial policy success rate $p_0$, HPT induces iterative improvement that satisfies:

$$p_{t+1} \geq p_t + \delta(p_t),$$

where in failure cases, supervised injection immediately boosts $p_t$, while in success cases the RL update yields monotonic amplification. By combining both, HPT retains a guaranteed upward drift in success probability.

## F DETAILS OF THE EXPERIMENTAL SETUP

For the benchmarks, AMC (Li et al., 2024) comprises problems drawn from the AMC12 2022 and AMC12 2023 examinations. ARC-c (Clark et al., 2018) is an open-domain reasoning benchmark and GPQA-Diamond (Rein et al., 2024) is a challenging and high-quality subset of the Graduate-Level Google-Proof Question Answering benchmark.

We set the training batch size to 128 and the maximum generation length to 8, 192 tokens, unless otherwise specified. We keep the system prompt and training datasets the same as LUFFY (Yan et al., 2025) and perform evaluation on DeepMath (He et al., 2025b). For other details that may not have been explicitly introduced, we have endeavored to follow previous works as closely as possible (Zhao et al., 2025; Zuo et al., 2025). All experiments were conducted on 8 x NVIDIA A800 80GB GPUs.

## G ABLATION STUDIES

For the ablation studies, § G.1 explores the role of off-policy RL, testing whether alternative strategies for utilizing offline data yield benefits. § G.2 presents a gate threshold ablation study.

### G.1 IMPACT OF OFF-POLICY RL

Table 5: Performance of different training paradigms to evaluate the impact of Off-policy RL. **SFT/ON** denotes SFT/On-policy (HPT), **OFF/ON** denotes Off-policy/On-policy, and **Mix/ON** denotes Mix-policy/On-policy.

| Name | AIME 24 | AIME 25 | AMC | MATH-500 | Minerva | Olympiad | Avg |
|---|---|---|---|---|---|---|---|
| OFF/ON | 16.6 | 11.8 | 47.3 | 76.2 | 35.3 | 41.6 | 38.1 |
| Mix/ON | **16.7** | 17.2 | 46.9 | 79.4 | 37.5 | 43.9 | 40.3 |
| SFT/ON | 16.6 | **17.8** | **51.0** | **81.0** | **37.5** | **47.3** | **41.9** |

In our previous work, we have only made preliminary attempts at unifying post-training by integrating RL with SFT. However, off-policy RL represents an important training paradigm that emphasizes leveraging the offline data. To this end, we further conduct experiments to investigate its influence and potential role.

We compare three different training paradigms: (1) SFT/On-policy, the model alternates between SFT and on-policy RL, which corresponds to the method we introduced above (HPT); (2) Off-policy/On-policy, the model alternates between off-policy RL and on-policy RL during training; and

(3) Mix-policy/On-policy, the model combines the loss from SFT and off-policy RL, and dynamically switches it with the on-policy RL objective. For the Mix setting, we performed hyperparameter search and found the optimal SFT/OFF weighting ratio to be $1/10$, i.e., the coefficients of the SFT loss and the off-policy loss are set to $0.1$ and $1.0$, respectively. We replicate the off-policy RL implementation described in LUFFY (Yan et al., 2025), and all experiments are conducted in the same settings to ensure fairness.

We evaluate the results of three methods on six math benchmarks. Table 5 presents results. Overall, the SFT/ON method achieves the best average performance (41.9), outperforming both Mix/ON (40.3) and OFF/ON (38.1). This suggests that off-policy RL may not be essential, as SFT already serves effectively as the training method of HPT for learning from offline data.

## G.2 GATE THRESHOLD ABLATION

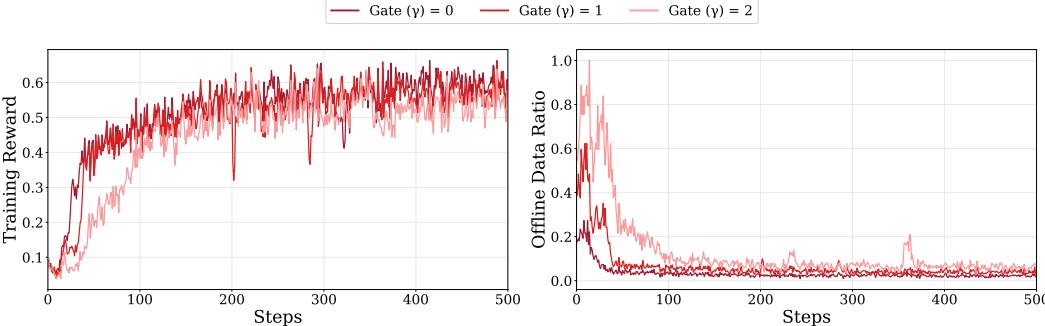

Figure 8: Training reward (left) and offline data ratio (right) comparisons across different gate settings on Qwen2.5-Math-1.5B.

In this section, we investigate the effect of different gate thresholds $\gamma$. A value of $\gamma = 0$ indicates that the model switches to SFT only when it fails all questions. Similarly, $\gamma = 1$ and $\gamma = 2$ correspond to settings where the model remains in on-policy reinforcement learning as long as it answers at least one or two out of eight questions correctly, respectively. To visualize the impact of the gating mechanism, we conduct experiments on the Qwen2.5-Math-1.5B under three different gate settings. As shown in Figure 8, we analyze the training dynamics by tracking the dynamics of rewards and the proportion of offline data utilized throughout training, thereby highlighting how different gate thresholds mediate the balance between leveraging offline demonstrations and incorporating online feedback. We observe that, under different gate thresholds, varying degrees of engagement with offline data–based SFT learning emerge. A larger gate threshold introduces a greater extent of SFT based on offline data, as expected.

Table 6: Performance of HPT with different switch gate $\gamma$ on Qwen2.5-Math-1.5B.

| Name | AIME 24 | AIME 25 | AMC | MATH-500 | Minerva | Olympiad | Avg |
|---|---|---|---|---|---|---|---|
| $\gamma = 2$ | 15.8 | 13.0 | 49.0 | 77.6 | 34.6 | 44.1 | 39.0 |
| $\gamma = 1$ | **18.1** | 14.2 | 46.0 | 75.4 | 35.7 | 42.5 | 38.7 |
| $\gamma = 0$ | 16.6 | **17.8** | **51.0** | **81.0** | **37.5** | **47.3** | **41.9** |

To further compare the performance across different gating strategies, we evaluate the three trained models on six benchmarks. Table 6 presents the results. Among the three configurations, $\gamma = 0$ achieves the best overall performance with an average score of 41.9, outperforming both $\gamma = 1$ (38.7) and $\gamma = 2$ (39.0). This observation suggests that simply incorporating more SFT does not necessarily lead to better outcomes. Instead, it is crucial to maintain a dynamic balance between the exploration of RL and the exploitation of SFT. The optimal degree of this gating mechanism should be adjusted according to the characteristics of the base model and the specific training data employed.

## H  THE USE OF LARGE LANGUAGE MODELS

We use large language models to polish our writing. Specifically, we employ ChatGPT (GPT-5 Thinking) to revise our initial manuscript section by section. The prompt we use is *I am writing an academic paper in English. Please polish the following draft so that it adheres to the conventions of academic writing.*

