# OpenReview forum: "Towards a Unified View of Large Language Model Post-Training"
_ICLR.cc/2026/Conference — ICLR 2026 Conference Desk Rejected Submission_

### Official Review · Reviewer_t8ra · 2025-10-27

**Soundness:** 3
**Presentation:** 3
**Contribution:** 3
**Rating:** 6
**Confidence:** 3

**Summary:**

This paper proposes the Unified Policy Gradient Estimator (UPGE) formulation, which unifies post-training practices for large language models, and introduces the Hybrid Post-Training (HPT) algorithm that adaptively selects between SFT and RL based on rollout quality. The UPGE comprises a Stabilization Mask, a Reference Policy Denominator, an Advantage Estimate, and a Likelihood Gradient. The HPT algorithm is designed to balance exploitation and exploration; specifically, it switches to SFT when the current performance on on-policy samples falls below a threshold, and to RL (in particular, Dr. GRPO) otherwise. The paper reports numerical experiments on mathematical reasoning benchmarks using Qwen and LLaMA models, showing that HPT outperforms SFT, GRPO, and mixed-policy baselines (e.g., LUFFY, SRFT). They observe notable gains on AIME-24 compared with LUFFY and SRFT, and demonstrate stronger Pass@k performance

**Strengths:**

The proposed UPGE offers a clear framework, making it easier to compare methods and systematically design new loss functions. HPT uses a simple binary, performance-gated switch that improves both accuracy and exploration depth (at least in the reported studies). Across multiple math-reasoning benchmarks and model families, the experimental results show consistent gains over strong baselines.

**Weaknesses:**

1. The empirical scope is narrow: experiments center on math/logic tasks with rule-based correctness verifiers. It’s unclear how HPT performs in domains without reliable automatic verifiers or with noisy, preference-based rewards.

2. The gating threshold is a critical hyperparameter. Although an ablation study is given in appendix and a general idea of ``maintain a dynamic balance between the exploration of RL and the exploitation of SFT'' is mentioned, the paper offers limited guidance on how to set it in practice.

3. It is unclear whether the reported improvement of the HPT truly comes from the gating or the extra information from the evaluation by the verifier.

**Questions:**

1. How should practitioners choose the gate threshold $\gamma$ in HPT? Furthermore, please explain the rationale behind your selected values of $\gamma$ for Qwen and LLAMA.

2. Please clarify the definition of $P$. You define  $P=\frac{1}{k} \sum_i v\left(\tau_i\right) \in[0,1]$ (a mean), yet later compare it to integer $\gamma \in\{0,1,2\}$. Are you actually thresholding the count $\sum_i v\left(\tau_i\right)$ (range $0 . . k$ )? If the mean is intended, how is an integer $\gamma$ interpreted? A precise statement will resolve this mismatch.

3. What verifiers are used for each benchmark and what are their known error rates? Please report verifier reliability in terms of false positives and false negatives.

4. Can the observed improvement of HPT be attributed to verifier reliability? It would be informative to conduct a small-scale label-noise stress test by injecting controlled noise into $v(\tau_i)$. How does HPT degrade as noise increases?

5. In the general mixed-loss form with coefficients $\alpha, \beta \geq 0$, but your main experiments set one to 0 and the other to 1 (binary gating). Have you evaluated fractional mixtures ( $0<\alpha, \beta<1$ )? A small ablation could show whether continuous mixing rivals or improves upon binary gating.

6. The numerical results focus on mathematical reasoning benchmarks. How does HPT perform in other tasks? Even a small-scale experiment would be helpful.

---

> ### Author Response · Authors · 2025-11-22
> **Response (1/2)**
>
> We greatly appreciate your insightful comments and suggestions.
>
> ## Response to Weakness 1 & Question 4:
>
> We add experiments to evaluate the performance of HPT under noisy rewards. In the original setting, the reliable verifier provides accurate rewards: a sampled answer receives a reward of 1 if it is correct and 0 if it is incorrect. We then inject noise on top of this reward by flipping it to the wrong value with probability p. We conduct two sets of experiments on Qwen2.5-Math-1.5B with p=0.01 and p=0.1, keeping all other settings unchanged. The results are as follows:
>
> | Noise Probability | AIME24 | AIME25 | AMC22+23 | MATH-500 | Minerva | Olympiad | AVG  |
> | ----------------- | ------ | ------ | -------- | -------- | ------- | -------- | ---- |
> | p=0               | 16.6   | 17.8   | 51       | 81       | 37.5    | 47.3     | 41.9 |
> | p=0.01            | 19.4   | 15.1   | 50.7     | 79.4     | 38.2    | 46.1     | 41.5 |
> | p=0.1             | 14.9   | 10.6   | 49.8     | 75.6     | 35.7    | 43.0     | 38.3 |
>
> When the noise probability is 0.01, the performance of HPT is almost unchanged, indicating that HPT is reasonably robust to small amounts of noise and can still perform well when the verifier is slightly unreliable. However, when the noise probability increases to 0.1, the performance of HPT degrades significantly, suggesting that its robustness to noisy rewards has an upper limit.
>
> ## Response to Weakness 2 & Question 1:
>
> We apologize for not providing more practical guidance on how to set this value. In practice, weaker models should use a larger gate value, because they need more guidance from demonstration data. If the model used by practitioners has comparable or better capability than Qwen2.5-Math-1.5B, we recommend setting the gate value to 0. For weaker models, such as LLaMA3.1-8B, the gate value can be set to 1/8 (1 out of 8 samples) or 2/8 (2 out of 8 samples). For even weaker models, for example undertrained models with fewer than 1B parameters, we expect that using a larger gate value would be beneficial.
>
> We choose the gate values for Qwen and LLaMA based on the heuristic that weaker models should use larger gate values, together with some preliminary experiments. On Qwen2.5-Math-1.5B, we find that setting the gate value to 0 yields the best performance. Since Qwen2.5-Math-7B is even stronger, we also set its gate value to 0. In contrast, LLaMA3.1-8B is weaker than Qwen2.5-Math-1.5B, so we set its gate value to 2 and observed performance that is substantially better than the baseline.
>
> ## Response to Weakness 3:
>
> All our RL-related baselines (RL, SFT-then-RL, SRFT, and LUFFY) use the same verifier. The extra information provided by the reliable verifier benefits all methods, not only ours. Therefore, we believe that the improvement of HPT mainly comes from appropriately switching between SFT and RL based on the model’s online performance, rather than from any additional information introduced by the verifier itself.
>
> ## Response to Question 2:
>
> Thank you for your careful checking! You are right that we did not describe $\gamma$ correctly in the latter part of the paper. There, all integer values of $\gamma$ were intended to denote the total number of correct answers among 8 samples. However, the correct definition of $\gamma$ should be the accuracy, rather than the raw count, which means $\gamma$ should take values in {$0,1/8,2/8, ...$} instead of {$0,1,2, ...$}. We apologize for this oversight and will correct it in the next version of our paper.
>
> ## Response to Question 3:
>
> For all benchmarks evaluated in the paper, we consistently use the verifier provided by DeepMath [1]. To assess the reliability of this verifier, we randomly sampled 20 trajectories from the inference outputs on AIME 2025 of the GRPO-trained checkpoint and 20 from the HPT-trained checkpoint, yielding 40 trajectories in total.
>
> We manually annotated the predicted answers for all 40 trajectories, and then used the DeepMath verifier to extract the model predictions. We found that every prediction extracted by the verifier matched the corresponding human-annotated result. Consequently, the measured error rate was 0, with 0 false positives and 0 false negatives.
>
> [1] DeepMath-103K: A Large-Scale, Challenging, Decontaminated, and Verifiable Mathematical Dataset for Advancing Reasoning

---

> ### Author Response · Authors · 2025-11-22
> **Response (2/2)**
>
> ## Response to Question 5:
>
> In the current implementation of HPT, α and β can only take binary values (0 or 1). They are discrete variables determined solely by the rollout performance feedback P. Making them continuous would require introducing additional components (e.g., a functional form or a small neural network).
>
> Below, we present an experiment designed to implement this setup. We use a function that maps the input rollout performance to the coefficients of RL and SFT (α and β). The values of this function under different rollout performance conditions are shown in the table below.
>
> | Rollout Performance | 0   | 1/8 | 2/8 | 3/8 | 4/8 | 5/8 | 6/8 | 7/8 | 1   |
> | ------------------- | --- | --- | --- | --- | --- | --- | --- | --- | --- |
> | α (for RL)          | 0.1 | 0.2 | 0.3 | 0.4 | 0.5 | 0.6 | 0.7 | 0.8 | 0.9 |
> | β (for SFT)         | 0.9 | 0.8 | 0.7 | 0.6 | 0.5 | 0.4 | 0.3 | 0.2 | 0.1 |
>
> Furthermore, we applied this function to implement the HPT, denoted as HPT (mix loss). We then compared it with the original HPT approach. The experimental results are shown in the table below.
>
> | Method         | AIME24 | AIME25 | AMC22+23 | MATH-500 | Minerva | Olympiad | Avg  |
> | -------------- | ------ | ------ | -------- | -------- | ------- | -------- | ---- |
> | HPT            | 16.6   | 17.8   | 51       | 81       | 37.5    | 47.3     | 41.9 |
> | HPT (mix loss) | 16.0   | 15.6   | 49.0     | 80.6     | 37.1    | 44.9     | 40.5 |
>
> It can be observed that the performance of HPT (mix loss) is comparable to that of the original HPT. This suggests that the effectiveness of HPT (mix loss) may be more sensitive to the choice of parameters of the gating mechanism.
>
> ## Response to Question 6:
>
> Your concerns are totally reasonable! However, due to constraints on both cost and time, we are currently unable to complete this experiment. We made an initial attempt, but soon realized that the overall workload and expense are far beyond what we had anticipated: it would require introducing a reward model, revising the gating strategy, tuning training hyperparameters, and constructing new training and evaluation datasets. We look forward to your understanding.
>
> However, we want to emphasize that HPT can in fact be transferred seamlessly to other domains. The essential RL differences across domains lie in the reward signal. For example, general domains may rely on a reward model, whereas math typically depend on a verifier to compute rule-based rewards. Therefore, when applying HPT to a different domain, the RL setup in that domain would naturally switch to its own reward system. In that case, the only required modification is to adjust the gating strategy to match the reward range of that system; no additional structural changes are needed.

---

> > ### Comment · Reviewer_t8ra · 2025-11-27
> >
> > I thank the authors for their thorough and constructive rebuttal. The additional experiments and clarifications address several of my earlier concerns. The response reflects the authors' effort to improve the submission.
> >
> > Some issues remain only partially resolved. The empirical scope is still largely limited to math-reasoning tasks with reliable verifiers. The gate threshold also remains a sensitive component of the method, and the practical guidance provided is still heuristic. These limitations do not change my overall assessment of the paper but do narrow the breadth of claims that can be supported.
> >
> > I therefore maintain my original rating.

---

> > > ### Author Response · Authors · 2025-11-28
> > >
> > > Thank you very much for your thoughtful engagement throughout the review process. We regret that the experiments could not be expanded beyond mathematical domains due to the constraints of the rebuttal period and cost. We would also like to indicate, as stated in our Response to Weakness 2 & Question 1, that all experiments can safely operate with the gate set to 0. This observation indicates that the threshold is in fact less sensitive than it may appear. Setting the gate to 0 never degrades performance: if a model is sufficiently strong and produces no rollout errors, HPT reduces exactly to RL; if the model is weaker and does exhibit such errors, SFT becomes necessary and a larger threshold is naturally beneficial. Thus, although tuning the gate is heuristic, our method guarantees that performance never decreases relative to the baseline. We therefore do not view this as true “sensitivity”. We hope this clarification fully alleviates your concern.
> > >
> > > We once again sincerely thank you for your comments. If you have any further questions or concerns, we would be happy to address them.

---

### Official Review · Reviewer_dbLE · 2025-10-27

**Soundness:** 3
**Presentation:** 3
**Contribution:** 3
**Rating:** 8
**Confidence:** 3

**Summary:**

The paper aims to unify supervised fine-tuning (SFT) and reinforcement learning (RL) fine-tuning of language models under a single framework. To this end, the paper introduces a unified gradient estimator (UPGE) that is derived from a common objective. Furthermore, the paper proposes a new retraining method named Hybrid pretreating (HPT) that uses a linear combination of SFT and RL. Empirical results are provided to demonstrate the strengths of HPT over SFT and GRPO (as an example of RL fine-tuning).

**Strengths:**

- The problem studied in the paper is highly relevant to area of post-training language models
- The motivated to study the problem is clearly stated in the paper. The common objective is sensible and the derivation is clear enough (although the appendix material could be simplified to hand-hold an inexperienced reader through the analytical steps). The gradient estimator appears to be correct.
- Empirical results are adequate to show the strengths of HPT. I do have a concern with the setup used in experimented noted in weakness
- Sufficient amount of ablations to show the strengths of HPT.

**Weaknesses:**

- HPT method uses a linear combination of SFT and RL losses in its objective. There is a new hyper parameter, \gamma, introduced in the paper that appears to require care while setting its value. No guidance is provided on how to set this value

- Empirical analysis uses Qwen2.5 and  LLAMA-3.1 models. The reader (including this reader) may wonder if the observations made with Qwen2.5 extends to Qwen3 and other models. In other words, why were these models chosen for experimentation and not other models? its not clear if the observations carry over to other models.

**Questions:**

(please see weakness for more info)

- How would a user set \gamma for a new model?

- Are there results available for other models? If so, this might help give the reader confidence about observations being general or applicable to more models than the ones used in the paper.

- The paper uses GRPO as a baseline but HPT itself uses Dr. GRPO. Would it make sense to report performance with Dr. GRPO method (as well)?

Overall, this is a good paper so I would accept given its strength with empirical and analytical analysis. My weakness observed above are shared in th hope that it might be useful for the authors to improve their draft.

---

> ### Author Response · Authors · 2025-11-22
> **Response**
>
> Thanks for your insightful and constructive comments and suggestions.
>
> ## Response to Weakness 1 & Question 1:
>
> We apologize for only providing a simple ablation study in Appendix G.2 without offering further practical guidance on how to set this value. In practice, weaker models should use a larger gate value, because they need more guidance from demonstration data. If the model used by practitioners has comparable or better capability than Qwen2.5-Math-1.5B, we recommend setting the gate value to 0. For weaker models, such as LLaMA3.1-8B, the gate value can be set to 1/8 (1 out of 8 samples) or 2/8 (2 out of 8 samples). For even weaker models, for example undertrained models with fewer than 1B parameters, we expect that using a larger gate value would be beneficial.
>
> ## Response to Weakness 2 & Question 2:
>
> We chose Qwen2.5-Math-7B, Qwen2.5-Math-1.5B, and LLaMA3.1-8B to follow the strongest baseline, LUFFY, which also evaluates only these three models. We fully agree that experiments on more models would further strengthen the credibility of our method. To this end, we additionally conducted experiments on Deepseek-Math-7B-base, comparing the base model, LUFFY, and our HPT on six benchmarks. We set the gate value to 2 (out of 8 samples), considering that Deepseek-Math-7B-base is slightly weaker than LLaMA-3.1-8B. All other training and evaluation settings follow those in the main paper. The results are shown in the following table.
>
> | Model                 | AIME24 | AIME25 | AMC22+23 | MATH-500 | Minerva | Olympiad | Avg   |
> | --------------------- | ------ | ------ | -------- | -------- | ------- | -------- | ----- |
> | Deepseek-Math-7b-Base | 0      | 0      | 1        | 4.2      | 1.1     | 1.2      | 1.25  |
> | LUFFY                 | 0.5    | 0.1    | 10.5     | 33.8     | 16.5    | 11.4     | 12.13 |
> | HPT                   | 4.3    | 0.5    | 14.6     | 43.4     | 20.6    | 13.2     | 16.1  |
>
> HPT still achieves the best performance on Deepseek-Math-7B-Base, which further demonstrates the robustness of our method across different backbone models.
>
> Because our training framework is built on verl 0.1.0 and vLLM 0.6.3, which do not support Qwen3, we are currently unable to evaluate HPT on Qwen3. We are upgrading our codebase and will run the corresponding Qwen3 experiments once this upgrade is completed.
>
> ## Response to Question 3:
>
> We sincerely apologize for the inaccurate wording in the paper. All of our experiments use the Dr.GRPO method, since the LUFFY paper has already shown that Dr.GRPO outperforms standard GRPO. We mentioned this choice in the Method section but did not clearly state it again in the Experiments section. We are sorry for this oversight and will correct it in the revised version of the paper, especially in the experimental part.

---

> > ### Comment · Reviewer_dbLE · 2025-11-26
> > **Thanks**
> >
> > I thank the author for their responses to my questions. I am satisfied with the responses made by the authors. I have also read other reviews and responses and can see that the authors have done their sincere best to respond to issues raised by my fellow reviewers.
> >
> > Overall, I see nothing that needs me to change my rating at this point. Keeping that value unchanged.

---

> > > ### Author Response · Authors · 2025-11-28
> > >
> > > Thank you for reviewing our work and for your recognition of our responses. We greatly appreciate your constructive feedback and will revise the manuscript accordingly.

---

### Official Review · Reviewer_HJxf · 2025-10-28

**Soundness:** 3
**Presentation:** 3
**Contribution:** 2
**Rating:** 6
**Confidence:** 4

**Summary:**

Formalizes existing reinforcement learning (RL) and supervised finetuning (SFT) methods as special cases of a more general Unified Policy Gradient Estimator (UPGE) framework. The framework allows accounting for differences in gradient masking (e.g., clipping in PPO), choice of reference policy, and advantage estimate. Then, the paper proposes a Hybrid Post-Training (HTP) procedure that chooses whether to perform SFT or RL updates based on whether the current accuracy of the model on a prompt is below or above a certain threshold. Empirical evaluations demonstrate that HPT surpasses the performance of several alternative post-training methods on mathematical reasoning and general knowledge datasets.

**Strengths:**

1. Relatively well-written and easy to follow.

2. The proposed HPT method is conceptually simple and shows promise compared to existing post-training baselines for tasks with verifiable rewards.

3. The empirical analysis of Section 4 is beneficial for understanding the inner-workings of HPT.

**Weaknesses:**

1. The significance of a unified perspective, such as the one proposed in Section 2, greatly depends on whether it allows characterizing similarities and differences between different instances. However, the unified framework is not really used in a meaningful way, which raises the question of why it is necessary or helpful. The proposed HPT method just boils down to either doing RL or SFT depending on an accuracy threshold for a prompt.

2. Related to the above, there is somewhat of a disconnect between the goal of providing a unified perspective for post-training and the proposed HPT, which is specific to tasks where you have verifiable binary rewards. These tasks can be quite narrow compared to the vast domains language models are trained for.

3. It is mentioned several times that different instantiations of UPGE exhibit different bias-variance tradeoffs. Yet, this claim is not well-substantiated in the paper, neither empirically nor theoretically. The main text refers to Appendix C, which lists differences between UPGE instantiations and hand-wavily provides arguments on bias and variance. But what are the biases and variance referred to in these arguments? The different instantiations optimize a different objective. Their optimal policies are different. How do you fairly compare bias and variance in this case? Also, given that the claims are more on the intuitive level and not formalized, it would be best to clearly state this to not mislead readers.

4. The paper does not report standard deviation across random seeds, or any other measure of statistical significance. So it is difficult to assess how consistent and substantial the improvements of HPT compared to baselines are.



Review Summary and Recommendation
---
Overall, the empirical results suggest that HPT is a simple yet competitive post-training method that combines the benefits of SFT and RL. However, the paper does not report any measure of statistical significance and the presentation and significance of the proposed unified framework can be substantially improved. As a result, I believe that the current paper falls on the borderline, where my assessment tends towards acceptance given that the authors can provide experiments verifying that the performance gains of HPT are consistent across random seeds.


Additional (More Minor) Comments
---

1. The paper refers to different post-training methods as optimizing the same "Common Objective". These statements, such as in lines 152-153, are often imprecise. Different methods do not optimize the same objective (e.g., this can be seen from the fact that the optimal policy with respect to them can differ), rather you can frame the different objectives as coming from the same objective family (defined by the "Common Objective"). It is important to clarify this point and avoid misleading claims, yet I do not believe this is a major issue since it can easily be fixed.

2. I recommend making the notation in the equation defining $grad_{Uni}$ more explicit so it is clearer what are the components involved (e.g., the prompt and response).

3. Something is off with Equation (4). If $\pi_{ref} = \pi_{\beta}$ then you do not get the correct gradient since there is a superfluous $\pi_{\beta} (\tau | q)$ (there is already one coming from the outer expectation). If $\pi_{ref} \neq \pi_{\beta}$ then also the gradient is incorrect since there is a missing division by $\pi_{ref}$.

4. The main text does not specify which data was used for training. This detail is found in the appendix, but I highly recommend putting it in Section 3.1 given its importance.

**Questions:**

--

---

> ### Author Response · Authors · 2025-11-22
> **Response (1/3)**
>
> Thank you for providing your valuable and constructive feedback.
>
> ## Response to Weakness 1:
>
> We fully agree with your view that UPGE should serve as a tool to better analyze the commonalities and differences among existing algorithms. We apologize for not clearly explaining the connection between UPGE and HPT in our original submission.
>
> We have added more detailed explanations and a deeper discussion in the General Response. In brief, we provide a more precise definition of bias and variance, and under the UPGE framework we can more clearly see that different algorithms all exhibit certain bias and variance. A further analysis shows that SFT is more suitable when the gap between the current model and the ideal optimal model is large, whereas RL is more suitable when this gap is small. At the same time, UPGE also indicates that these two algorithms can be used within the same training stage without causing conflicts. Therefore, a better strategy is to dynamically choose SFT or RL based on the model’s performance on the current samples, which is exactly what our HPT algorithm does. We will incorporate these clarifications into the next version of our paper to more clearly highlight the important role of UPGE.
>
> ## Response to Weakness 2:
>
> Our goal is to provide a unified perspective that allows us to better analyze existing methods and to design improved ones. Essentially, HPT dynamically chooses between SFT and RL based on the current model’s performance (i.e., the reward). This means that HPT can be applied to any task where rewards are available (e.g., RLHF), and is not limited to tasks with verifiable binary rewards. We conduct experiments on mathematical reasoning tasks in order to follow our main baseline LUFFY and thus enable a fair comparison.
>
> However, we acknowledge that the presentation in our original submission should be improved to more clearly clarify the connection between UPGE and HPT. We also acknowledge that HPT indeed cannot be used on tasks where rewards are unavailable (e.g., pure SFT), which is an inherent limitation of HPT.
>
> ## Response to Weakness 3:
>
> We fully agree with your point that the biases and variance need to be defined more clearly. In our setting, the biases and variance refer to the bias and variance of the optimization objective induced by each algorithm, relative to the ideal optimal objective. In the General Response, we have added explicit definitions of bias and variance, and we analyze in detail their sources as well as their concrete forms under different algorithms. We will incorporate these clarifications into the next version of our paper.

---

> ### Author Response · Authors · 2025-11-22
> **Response (2/3)**
>
> ## Response to Weakness 4:
>
> For evaluation datasets with a small number of examples (AIME24, AIME25, and AMC), we already report the avg@32 metric to mitigate randomness. However, we agree with your point that randomness still exists. To further reduce the impact of randomness, we performed repeated experiments from both the evaluation and training stages.
>
> For evaluation, we re-run all the evaluations we conducted in Tables 2 and 3 of the original paper. We set the random seeds to 42, 21, and 63, and perform three independent evaluation runs. The means and standard deviations of these three runs are reported in the two tables below, where the number in parentheses in each cell denotes the standard deviation. We can see that, after accounting for randomness, our method HPT still achieves the best performance.
>
> Table 2 (Qwen2.5-Math-7B)：
> | Model    | AIME24     | AIME25     | AMC        | MATH-500   | Minerva    | Olympiad   | `Avg`        | ARC-c      | GPQA       | `Avg`        |
> | -------- | ---------- | ---------- | ---------- | ---------- | ---------- | ---------- | ---------- | ---------- | ---------- | ---------- |
> | Base     | 11.6 (0.6) | 4.8 (0.3)  | 33.1 (0.1) | 47.3 (2.6) | 9.3 (0.7)  | 13.8 (0.5) | `20.0 (0.5)` | 29.8 (0.9) | 28.0 (0.5) | `28.9 (0.6)` |
> | SFT      | 23.1 (2.1) | 22.3 (5.4) | 55.9 (0.7) | 83.7 (0.5) | 34.8 (1.7) | 43.8 (1.1) | `43.9 (1.1)` | 68.9 (1.1) | 25.6 (2.1) | `47.2 (1.2)` |
> | RL       | 19.5 (0.3) | 13.6 (0.7) | 59.3 (0.3) | 81.8 (0.1) | 38.5 (0.4) | 45.8 (0.6) | `43.1 (0.3)` | 80.9 (0.3) | 36.2 (2.7) | `58.5 (1.3)` |
> | SFT → RL | 24.9 (0.9) | 20.6 (2.2) | 62.3 (1.2) | 85.3 (0.8) | 38.1 (1.1) | 47.4 (0.9) | `46.4 (0.2)` | 66.9 (0.6) | 31.1 (0.5) | `49.0 (0.2)` |
> | LUFFY    | 26.4 (0.8) | 21.6 (0.6) | 66.5 (0.2) | 88.0 (0.3) | 42.0 (0.2) | 54.4 (0.5) | `49.8 (0.1)` | 80.5 (1.0) | 38.4 (2.2) | `59.4 (1.6)` |
> | SRFT     | 19.3 (0.7) | 15.8 (0.5) | 56.0 (0.1) | 83.3 (0.4) | 42.6 (0.3) | 48.3 (0.7) | `44.2 (0.1)` | 80.1 (0.4) | 36.2 (2.1) | `58.2 (0.9)` |
> | HPT      | 32.4 (0.4) | 22.3 (0.4) | 69.2 (0.2) | 89.5 (1.0) | 46.6 (0.8) | 57.1 (0.5) | `52.9 (0.3)` | 81.5 (0.2) | 39.4 (1.4) | `60.5 (0.8)` |
>
> Table 2 (Qwen2.5-Math-1.5B & Llama3.1-8B)：
> | Model             | AIME24     | AIME25     | AMC        | MATH-500   | Minerva    | Olympiad   | Avg        |
> | ----------------- | ---------- | ---------- | ---------- | ---------- | ---------- | ---------- | ---------- |
> | Qwen2.5-Math-1.5B |            |            |            |            |            |            |            |
> | Base              | 5.0 (1.6)  | 3.9 (1.6)  | 24.6 (0.6) | 34.3 (2.2) | 11.3 (1.2) | 16.6 (0.3) | 16.0 (0.4) |
> | SFT               | 15.2 (1.4) | 16.7 (0.7) | 44.7 (0.5) | 77.1 (1.0) | 31.4 (1.8) | 36.5 (1.2) | 36.9 (0.6) |
> | RL                | 12.0 (0.4) | 8.3 (0.1)  | 44.1 (0.2) | 70.7 (0.5) | 32.5 (1.7) | 35.0 (0.2) | 33.8 (0.4) |
> | LUFFY             | 12.4 (1.2) | 7.8 (1.3)  | 44.5 (1.8) | 75.3 (0.9) | 28.9 (2.0) | 39.2 (0.6) | 34.7 (0.5) |
> | HPT               | 16.8 (0.3) | 17.2 (0.4) | 51.1 (0.1) | 80.3 (0.5) | 37.2 (0.9) | 47.4 (0.3) | 41.7 (0.2) |
> | Llama3.1-8B       |            |            |            |            |            |            |            |
> | Base              | 0.4 (0.1)  | 0.2 (0.0)  | 4.8 (0.1)  | 13.5 (0.6) | 5.0 (0.3)  | 3.4 (0.6)  | 4.5 (0.1)  |
> | HPT               | 2.0 (0.1)  | 1.5 (0.2)  | 19.0 (0.3) | 47.7 (0.1) | 18.4 (0.6) | 19.5 (0.7) | 18.0 (0.1) |
>
> For training, we performed two additional HPT training on Qwen2.5-Math-1.5B with different random seeds (21, 63). Together with the original run, this yields three checkpoints. We evaluate each checkpoint three times with different random seeds and then compute the mean and standard deviation. The results are as follows:
>
> | Train seed | AIME24     | AIME25     | AMC22+23   | MATH-500   | Minerva    | Olympiad   | AVG        |
> | ---------- | ---------- | ---------- | ---------- | ---------- | ---------- | ---------- | ---------- |
> | 42         | 16.8 (0.3) | 17.2 (0.4) | 51.1 (0.1) | 80.3 (0.5) | 37.2 (0.9) | 47.4 (0.3) | 41.7 (0.2) |
> | 21         | 16.9 (0.3) | 16.7 (1.2) | 50.8 (0.1) | 80.6 (0.4) | 38.5 (0.6) | 46.3 (0.8) | 41.6 (0.3) |
> | 63         | 17.2 (0.3) | 16.1 (0.5) | 52.2 (2.9) | 80.3 (0.2) | 37.5 (1.3) | 44.2 (0.2) | 41.3 (0.4) |
>
> We observe that the mean performance across checkpoints trained with different seeds fluctuates only mildly, which further demonstrates that the effectiveness of HPT is robust and only weakly affected by random factors.

---

> ### Author Response · Authors · 2025-11-22
> **Response (3/3)**
>
> ## Response to Additional Comment 1:
>
> We agree with your comment. There are indeed some minor issues in how we introduce and define the “Common Objective”, and we will correct and clarify them in the next revision of the paper.
>
> ## Response to Additional Comment 2:
>
> The formula defining $grad_{Uni}$ is indeed too concise. Although our intention was to highlight the four constituent components, this definition is not explicit enough. We will revise it to a more detailed version to make all the involved components explicit.
>
> ## Response to Additional Comment 3:
>
> We are sorry for our mistake. Equation (4) is indeed incorrect. We introduced an extra indicator term $\mathbf{1}$ {$\pi_{ref} = \pi_{\beta} $}. Once this term is removed, the equation becomes mathematically sound, and we provide a detailed derivation in https://postimg.cc/m19cw87n. We originally added $\mathbf{1}$ {$\pi_{ref} = \pi_{\beta} $} to more clearly distinguish between the SFT and RL cases, but this indeed led to a lack of mathematical rigor. We apologize again for this oversight.
>
> ## Response to Additional Comment 4:
> We agree with your suggestion, and we will move the description of the training datasets into the main text.

---

> > ### Comment · Reviewer_HJxf · 2025-11-23
> >
> > Thank you for the response. I have read it and the other reviews carefully. I appreciate the additional experiments, which report standard deviation across random seeds for sampling and training. It is worth noting that, while for the Qwen2.5-Math-1.5B and Llama-3.1-8B models the gap between HPT and the remaining methods seems significant, it is much less the case for the Qwen2.5-Math-7B model, where both RL and LUFFY achieve roughly similar performance when accounting for standard deviation.
> >
> > Furthermore, weaknesses 1, 2, and 3 in my original review largely stand. In particular, the discussion around bias-variance tradeoffs of different UPGE gradient estimators is quite simplistic. The claims made regarding why certain methods should have a higher bias or variance are not meaningfully substantiated. It is therefore unclear to me the extent to which they motivate the use of HPT or, more broadly, aid the contributions of the paper. I strongly recommend explicitly stating that the discussion around bias-variance tradeoffs is on an intuitive level to not mislead readers.
> >
> > Given the above, I maintain my initial weakly positive assessment of the paper.

---

> > > ### Author Response · Authors · 2025-11-28
> > >
> > > We sincerely thank you for your further valuable comments.
> > >
> > > For the Qwen2.5-Math-7B model, HPT achieves a 3.1 point improvement over LUFFY (49.8 → 52.9) in the average performance across the 6 in-distribution tasks, which we believe more directly reflects the relative effectiveness of different methods than the out-of-distribution tasks. Compared with RL, HPT achieves a 9.8 point improvement (43.1 → 52.9). We have revised the table in Response (2/3) to more clearly present the scores on in-distribution and out-of-distribution tasks.
> > >
> > > We fully agree with your point that our discussion of the bias–variance tradeoffs lacks rigorous quantitative analysis, and we will make it explicit in the paper that most of our discussion on bias–variance tradeoffs is at an intuitive level. Nevertheless, we believe that our analysis of the sources of bias and variance, as well as the connection between the common objective and classical RL, is still insightful and provides useful guidance for the design of HPT. We will retain these parts that we consider valuable in the revised version of the paper, and we would very much welcome any further discussion with you.
> > >
> > > We once again sincerely thank you for your comments, and we apologize for our late response. If you have any further questions or concerns, we would be happy to address them.

---

### Official Review · Reviewer_tTEQ · 2025-10-31

**Soundness:** 2
**Presentation:** 3
**Contribution:** 2
**Rating:** 6
**Confidence:** 4

**Summary:**

The paper proposes UPGE - a unified policy gradient estimator for post-training algorithms like RL and sft. Inspired by the analysis in UPGE, the author aimed to combine RL with sft using a dynamic gating mechanism based on the model's performance on the question.

**Strengths:**

* Good paper writing, clear motivation
* The proposed HPT yielded noticeable improvement against RL or SFT

**Weaknesses:**

* It’s not clear to me how the unified gradient estimator connects to the proposed HPT. Sections 2.1 and 2.2 do not clearly bridge the concepts to the HPT design described in Section 2.3.
* Specifically, following the previous point, in order to unify RL and SFT under a common objective, Section 2.2 introduces a non-conventional definition of the post-training optimization target by replacing the original \pi_{ref} with the behavior policy in the RL context. I think the paper should clarify this setting in more detail.
* Is there any ablation or analysis on how changes in the gating threshold affect the training dynamics?

**Questions:**

Just a comment that related to the weaknesses mentioned earlier. The writing of the paper gives me the impression that the authors first built something that works (combining RL and SFT dynamically) and then retroactively proposed a theoretical framework to justify it. The presentation was a bit confusing; I only understood the UPGE part after reading the methodology section and then revisiting it.

---

> ### Author Response · Authors · 2025-11-22
> **Response**
>
> We greatly appreciate your insightful comments and suggestions.
>
> ## Response to Weakness 1:
>
> We apologize for not clearly explaining the connection between UPGE and HPT in our original submission. We now provide a more detailed explanation and deeper discussion in the General Response. Briefly, under the UPGE framework, we can more clearly see that different algorithms exhibit specific forms of bias and variance. A more in-depth analysis shows that SFT is more suitable when the gap between the current model and the ideal optimal model is large, whereas RL is more appropriate when this gap is small. UPGE further shows that these two algorithms can be used within the same training stage without causing conflicts. Therefore, a better strategy is to dynamically choose between SFT and RL based on the model’s performance on the current samples, which is exactly what our HPT algorithm does. We will incorporate these clarifications into the next version of our paper to more clearly present the connection between UPGE and HPT.
>
> ## Response to Weakness 2:
>
> We agree that the explanation here is not sufficiently detailed. We will address your concerns from two perspectives and will incorporate the corresponding revisions into the next version of the paper.
>
> First, $\pi_{ref}$ is not being replaced. We speculate that by “original $\pi_{ref}$” you are referring to the reference policy (typically the initial model) used in the KL loss $\text{KL}(\pi_{\theta} || \pi_{ref})$ in RL, which prevents the policy from drifting too far from its starting point during training. We are not using the $\text{KL}(\pi_{\beta} || \pi_{\theta})$ in Eq. (1), to replace this KL loss $\text{KL}(\pi_{\theta} || \pi_{ref})$. We temporarily omitted the KL term $\text{KL}(\pi_{\theta} || \pi_{ref})$ in Eq. (1) for simplicity. We provide a detailed discussion of the case where the $\text{KL}(\pi_{\theta} || \pi_{ref})$ is included in Appendix D.2, and the last equation on page 19 of our paper shows the resulting form of the Common Objective when this term is added. This KL term $\text{KL}(\pi_{\theta} || \pi_{ref})$ can be absorbed into the advantage function.
>
> Second, the introduction of $\text{KL}(\pi_{\beta} || \pi_{\theta})$ is intended to use the distribution of demonstration data (or the sampling policy) to approximate the ideal optimal policy, which corresponds to the first equation we mentioned in the General Response. This is precisely the approximation we make when transferring classical RL to the language model setting.
>
> ## Response to Weakness 3:
>
> We provide a detailed analysis of the training dynamics under different gating thresholds in Appendix G.2. We observe that, under different gate thresholds, varying degrees of engagement with offline-data–based SFT emerge. A larger gate threshold induces a greater amount of SFT based on offline data. If further ablation studies are needed, we would be happy to provide them.
>
> ## Response to Questions:
>
> We agree with your comment that the presentation in our original submission may lead readers to misunderstand the causal relationship between UPGE and HPT. Our goal is to unify existing algorithms within a single framework and, on this basis, propose a better algorithm. However, due to limitations in our writing, we did not elaborate on the connection between UPGE and HPT in sufficient detail, and we apologize for the resulting confusion. In the General Response, we now describe in detail how we use UPGE to analyze existing algorithms and ultimately arrive at our HPT algorithm. We will include these revisions in the next version of our paper.

---

### Official Review · Reviewer_PwLU · 2025-11-01

**Soundness:** 3
**Presentation:** 3
**Contribution:** 3
**Rating:** 6
**Confidence:** 4

**Summary:**

This paper proposes the Unified Policy Gradient Estimator (UPGE), a theoretical framework that unifies various LLM post-training algorithms including SFT and RL methods through their loss gradient form. The authors demonstrate that these seemingly disparate approaches optimize a common objective under different data distribution assumptions and bias-variance tradeoffs. Building on this unified view, the paper introduces Hybrid Post-Training (HPT), an algorithm that dynamically switches between SFT and RL based on real-time performance feedback. Extensive experiments on mathematical reasoning benchmarks show that HPT consistently outperforms strong baselines including SFT, GRPO, LUFFY, and SRFT across models of varying scales (Qwen2.5-Math-1.5B/7B, LLaMA3.1-8B).

**Strengths:**

1. The paper makes a strong theoretical contribution by deriving UPGE from a common objective (Eq. 1), showing that SFT and RL are not contradictory but complementary learning signals.

2. The proposed HPT algorithm is well-motivated by the unified theoretical framework. The dynamic switching mechanism based on performance feedback (Eq. 6-8) is practical, avoiding the need for manual tuning of mixing ratios as in prior work like LUFFY.

3. The experimental evaluation is comprehensive, covering multiple model scales (1.5B, 7B, 8B) and families (Qwen, LLaMA), six mathematical reasoning benchmarks and two out-of-distribution tasks.

4. The paper is generally well-written with clear motivation and organization.

**Weaknesses:**

1. The claim that the multi-stage (SFT->RL) process is notoriously resource-intensive seems unreasonable, since SFT requires far fewer computational resources compared to RL.

2. The paper mentions that "SFT memorizes, RL generalizes" in related work (Chu et al., 2025) but does not systematically investigate the memorization-generalization tradeoff in HPT. Does HPT's dynamic switching lead to better generalization on out-of-distribution tasks? The limited evaluation on only two OOD tasks (ARC-c, GPQA) is insufficient to draw strong conclusions.

3. The paper does not provide sufficient analysis on computational costs. While HPT avoids the two-stage SFT→GRPO pipeline, it still requires generating multiple rollouts for each question to compute performance feedback. A detailed comparison of training time, GPU memory, and total compute budget against baselines would strengthen the practical applicability claims.

4. The HPT algorithm relies on performance feedback coefficients α and β, as well as the switch threshold γ, which significantly impact model performance. However, the paper provides only limited empirical settings (e.g., γ=0 or 2) without a systematic sensitivity analysis or guidance on adjustment.

**Questions:**

N/A

---

> ### Author Response · Authors · 2025-11-22
> **Response (1/3)**
>
> Thanks for your insightful and constructive comments and suggestions.
>
> ## Response to Weakness 1:
>
> We fully understand your concerns, and we clarify that the resource intensity does not stem from the SFT stage alone. Instead, it arises from the fact that SFT and RL are cascaded within a curriculum-learning pipeline. In practice, even though SFT is relatively lightweight compared to RL, the overall tuning workflow becomes resource-heavy for the following reasons:
> 1. Iterative feedback loops: It is common for the RL results to reveal issues that require redesigning or re-training the SFT stage, such as adjusting data curation, objective formulation, or alignment strategies. Each iteration requires re-running RL, which is the most expensive part.
> 2. Coupled, explosion-prone tuning: Because SFT and RL are tightly interdependent, the effect of any SFT change can only be judged after full downstream RL runs. This coupling makes pipeline tuning combinatorial, where even small SFT adjustments trigger many RL trials, driving up cost, time, and operational complexity.
>
> Moreover, in practice, an SFT-trained model (especially based on long CoT) can greatly increase the time required for subsequent RL rollouts. In contrast, HPT dynamically and selectively incorporates SFT, thereby avoiding this issue. So although SFT alone is not resource-intensive, the multi-stage dependency structure makes the entire SFT→RL pipeline significantly more demanding than single-stage methods.
>
> ## Response to Weakness 2:
>
> We additionally evaluate on three out-of-distribution tasks (MMLU, MMLU-Pro, and LiveCodeBench) to assess the generalization ability of HPT. We report avg@1 on MMLU and MMLU-Pro and avg@16 on LiveCodeBench, while keeping the rest of the evaluation setup identical to that in the main paper. The results on all five out-of-distribution tasks are summarized in the table below:
>
> | Model           | ARC-c    | GPQA     | MMLU     | MMLU-Pro | LiveCodeBench | Avg       |
> | --------------- | -------- | -------- | -------- | -------- | ------------- | --------- |
> | Qwen2.5-Math-7B | 30.9     | 28.3     | 44.8     | 25.4     | 11.3          | 28.14     |
> | SFT             | 67.4     | 25.3     | 58.8     | 47.2     | 9.1           | 41.56     |
> | RL              | 81.2     | 36.4     | 60.7     | 40.7     | 6.5           | 45.1      |
> | SFT → RL        | 67.7     | 30.8     | 49.5     | 22.0     | 9.0           | 35.8      |
> | LUFFY           | 80.8     | 39.4     | **63.8** | 52.0     | 12.4          | 49.68     |
> | SRFT            | 80.5     | 36.8     | 58.1     | 44.0     | 15.2          | 46.92     |
> | HPT             | **81.6** | **42.9** | 63.6     | **52.5** | **18.3**      | **51.78** |
>
> HPT achieves the comparative or best performance on all five benchmarks, which demonstrates that it leads to better generalization on OOD tasks. This may be because our method introduces RL at an optimal stage of training, thereby endowing the model with strong generalization ability.

---

> ### Author Response · Authors · 2025-11-22
> **Response (2/3)**
>
> ## Response to Weakness 3:
>
> In theory, the computational cost of HPT is essentially the same as that of vanilla GRPO. The step of generating multiple rollouts in HPT is also an indispensable part of GRPO. The training pipeline of HPT is “sampling, obtaining rewards, deciding whether to perform SFT or RL, computing the loss, and updating,” whereas the training pipeline of GRPO is “sampling, obtaining rewards, computing the loss, and updating.” The only differences lie in the additional decision of whether to perform SFT or RL and the specific loss formulation, whose extra computational overhead is negligible. Therefore, if we assume that their effective sampling lengths are the same, the computational cost of HPT is effectively identical to that of vanilla GRPO.
>
> However, in our experiments we observe that, for the SFT→GRPO setting, because SFT is performed beforehand, the rollout length in the subsequent GRPO phase becomes longer, which substantially increases the wall-clock time of GRPO. On the same 8-GPU H20 machine, we ran 50 steps of HPT and “50 steps of SFT + 50 steps of GRPO,” and recorded GPU memory usage and training time per step. Under comparable GPU memory usage, the training time of SFT→GRPO is significantly higher than that of HPT.
>
> |          | GPU Memory (GiB) | Training Time Per Step （second） |
> | -------- | ---------------- | ------------------------------- |
> | SFT → RL | 87.22            | 190.66                          |
> | HPT      | 87.16            | 414.63                          |

---

> ### Author Response · Authors · 2025-11-22
> **Response (3/3)**
>
> ## Response to Weakness 4:
>
> We apologize for the lack of clarity in the manuscript that led to your misunderstanding. We provide detailed clarifications below:
>
> 1. Existing Ablation Studies
>
> - First, we would like to clarify that α and β are not tunable parameters of HPT. They are outputs of the gate mechanism in HPT. Their values depend not only on the specific question but also vary dynamically throughout the entire training process. Moreover, we find that treating α and β as binary variables already offers a sufficiently simple design, which was part of our original motivation, to minimize potential confounding effects introduced by additional tunable parameters.
> - Second, regarding γ, we have included the corresponding ablation study in Appendix G. We apologize for not referencing it explicitly in the main text, and we hope the experiments presented there address your concerns. If further ablation studies are needed, we would be happy to provide them.
> - Finally, concerning the guidance on adjustment: since α and β have not tunable in their current binary form, we do not elaborate further on them. As for γ, it can be safely set to 0 in all experiments. This setting is broadly applicable because γ = 0 reflects scenarios in which RL cannot obtain correct answers through rollout. In such cases, incorporating SFT based on offline trajectories is both reasonable and the only remaining safe choice in practice.
>
> 2. Further Ablation Studies
>
> Overall, the only tunable component in HPT is the gate mechanism itself. The gate mechanism takes the performance feedback P as input and produces α and β as outputs. The tunable aspects of the gate mechanism mainly include two parts:
> - Mechanism pipeline: for example, using a threshold-based binary gate or defining a functional mapping for the gate.
> - Mechanism parameters: such as the current gate threshold γ, among others.
>
> The experiments in Appendix G can be viewed as an ablation on the parameters. To fully resolve your concerns, we also conducted an ablation on the mechanism pipeline. In the current implementation of HPT, α and β can only take binary values (0 or 1). They are discrete variables determined solely by the rollout performance feedback P. Making them continuous would require introducing additional components (e.g., a functional form or a small neural network).
>
> Below, we present an experiment designed to implement this setup. We use a function that maps the input rollout performance to the coefficients of RL and SFT (α and β). The values of this function under different rollout performance conditions are shown in the table below.
>
> | Rollout Performance | 0   | 1/8 | 2/8 | 3/8 | 4/8 | 5/8 | 6/8 | 7/8 | 1   |
> | ------------------- | --- | --- | --- | --- | --- | --- | --- | --- | --- |
> | α (for RL)          | 0.1 | 0.2 | 0.3 | 0.4 | 0.5 | 0.6 | 0.7 | 0.8 | 0.9 |
> | β (for SFT)         | 0.9 | 0.8 | 0.7 | 0.6 | 0.5 | 0.4 | 0.3 | 0.2 | 0.1 |
>
> Furthermore, we applied this function to implement the HPT, denoted as HPT (mix loss). We then compared it with the original HPT approach. The experimental results are shown in the table below.
>
> | Method         | AIME24 | AIME25 | AMC22+23 | MATH-500 | Minerva | Olympiad | Avg  |
> | -------------- | ------ | ------ | -------- | -------- | ------- | -------- | ---- |
> | HPT            | 16.6   | 17.8   | 51       | 81       | 37.5    | 47.3     | 41.9 |
> | HPT (mix loss) | 16.0   | 15.6   | 49.0     | 80.6     | 37.1    | 44.9     | 40.5 |
>
> It can be observed that the performance of HPT (mix loss) is comparable to that of the original HPT. But the effectiveness of HPT (mix loss) may be more sensitive to the choice of parameters of the gating mechanism.

---

### Author Response · Authors · 2025-11-22
**General Response to Reviewers: Connections, Motivations, and Formal Definitions (1/3)**

We thank all the reviewers for their careful evaluations and thoughtful comments. Notably, almost all reviewers mentioned that our work is well written, easy-to-follow, and well motivated. We sincerely acknowledge the general positivity of all reviewers. However, multiple reviewers have pointed out that there currently exists a disconnection between our theoretical motivations regarding the unified perspective and the proposed HPT algorithm.

On behalf of all authors, we wholeheartedly agree that in our original submission, the writing can be greatly improved, some technical details are necessary for our narrative, and some discussions/statements were missing in our main paper as well as our appendix. We thank the reviewers for their suggestions.

Below, we provide a compilation of subsections that are previously missing from the original submission, including:
- Connections between the Common Objective and classical RL literature.
- The source of bias and variance in LLM post-training
- Connection between UPGE and HPT: Why HPT selects a better bias-variance tradeoff w/ a numerical optimization example

These additions will also be added to our revised draft, to appear in either the main paper or the appendix. We hope the reviewers find these additions welcoming.

---

> ### Author Response · Authors · 2025-11-22
> **General Response to Reviewers: Connections, Motivations, and Formal Definitions (2/3)**
>
> # Locating Bias and Variance from the Common Objective
>
> Given the definition of the Common Objective in Eq. (1) (of the original paper), here we provide the source of bias and variance in the existing instances of the UPGE from a technical perspective.
>
> ## Connection with Classical RL
>
> We first assume the existence of an **optimal policy**, denoted as $\pi^*$, which describes the ideal response from the model given any request. Suppose that we can obtain unlimited trajectories from the optimal policy, then we can write the Common Objective as a classical KL-regularized reinforcement learning objective [https://web.stanford.edu/class/psych209/Readings/SuttonBartoIPRLBook2ndEd.pdf, section 3, equation 3.14] such that
>
> $$
> \mathcal{J}_\mu^\*(\theta)
> = \mathbb{E}\_{s \sim \pi^\*} \big[\pi\_{\theta}(a\_t|s) Q^\*(s, a\_t) - \mu \, KL(\pi^\* \|\| \pi\_{\theta})\big],
> \tag{1}
> $$
>
> where $ Q^\*(s, a) = \mathbb{E}\_{s_t, a_t \sim \pi^\*} \Big[\sum\_{t} r(s_t, a_t)\\big|s_0 = s, a_0 = a\Big] \tag{N}$
>
> Notably, $\mu$ can take an arbitrary value since the solution of the Common Objective coincides with $\pi^*$.
>
> In the context of language models, we treat the current state as $s_t := \\{q, \tau_{<t}\\}$ and the action as the current token $a_t := \tau_t.$ Therefore, the Common Objective can be written as
>
> $$
> \mathcal{J}_\mu^\*(\theta)
> = \mathbb{E}\_{q,\tau\_{<t} \sim \pi^\*} \Big[
> \pi\_{\theta}(\tau_t \mid q,\tau\_{<t}) \,
> \mathbb{E}\_{\tau\_{>t}\sim\pi^\*} \big[r(\tau\_{<t}, \tau_t, \tau\_{>t})\big] - \mu KL(\pi^\* \|\| \pi\_{\theta})
> \Big].
> \tag{2}
> $$
>
> This objective refers to the optimization of the current policy on any single token output, following the token-trajectory distribution of $\pi^\*$. In this case, the gradient of Eq. (2), $\nabla_\theta \mathcal{J}_\mu^*(\theta)$, can be seen as a policy gradient with no bias and no variance, which we denote as the **oracle gradient**.
>
> ## Source of Bias and Variance
>
> Due to the vast action space and immense sampling complexity of language models, and that the optimal policy is practically unattainable, to design a computationally tractable algorithm, a number of approximations are required. In practice, all UPGEs are calculated given finite samples from some behavior policy $\pi_{\beta}$. This procedure induces **bias and variance** into the gradient estimator with respect to the oracle gradient.
> From a statistical perspective, the source of variance is the finite number of samples over the entire trajectory distribution, which also includes additional importance sampling operations. Bias comes from multiple sources. Notably, the difference between the behavior policy and the optimal policy is a major source of bias, and the selectivity in sampling introduces another source of bias.
>
> Here, we provide a brief discussion on the instances of bias and variance in some existing UPGE components. In Eq. (5) of the original paper, we can see that the reference policy denotes the selection of a behavior policy. Therefore, the selection of such polices denotes a trade-off between bias and variance.
>
> - For the case of **SFT**, a demonstration dataset is used to approximate the optimal policy. In this case, the bias comes from the difference between demonstrated answers and the optimal answers, whereas the variance comes from the relatively low number of correct answer samples compared to RL approaches.
> - On the other hand, **GRPO** uses $\pi_{\theta_{old}}$ as the reference policy. This sheds light onto the dependence of GRPO on the base model performance: The bias of the GRPO gradient is greatly dependent on the similarity of the rollout policy and the optimal policy; the gradient estimation is less biased with a high-quality rollout policy and more biased with a low-quality rollout policy.
>
> We refer to Appendix C for a more detailed discussion on the bias and variance of other existing components, i.e. Advantage estimate and Stability Mask.

---

> ### Author Response · Authors · 2025-11-22
> **General Response to Reviewers: Connections, Motivations, and Formal Definitions (3/3)**
>
> # Selecting the Best Bias-Variance Tradeoff for Post-Training
>
> From the previous analysis, we can conclude that not only do the different components of UPGE induce different bias-variance tradeoffs, but more importantly, these tradeoffs also vary during the training process and are dependent on the current performance of the model. We conjecture that SGD exhibits moderate bias (stemming from demonstration data) and relatively high variance (due to limited samples), whereas GRPO-style RL algorithms exhibit lower variance but a performance-dependent bias. Therefore, it can be expected that between the two algorithms, SFT exhibits a better bias-variance tradeoff towards the start of the training process, and GRPO is favorable towards the end, with model performance serving as an indication.
>
> We provide a numerical example for an optimization task with a similar dynamical bias-variance tradeoff. We note that although the LLM post-training task is much more complicated with strong non-convex and non-smooth properties, this example can still serve as an intuition to guide algorithmic designs for our proposed HPT in Section 4.
>
> In this experiment, we consider a classic least-squares minimization problem, $f(\theta) = \|A \theta - b\|^2,$ where all entries of $A$ and $b$ follow a normal distribution. In this problem, the oracle gradient can be easily calculated, which we denote as $\text{grad}\_{oracle}(\theta) = \nabla_\theta f(\theta).$
>
> We construct two different gradient estimators, $\hat{\text{grad}}\_{SFT}$ and $\hat{\text{grad}}\_{GRPO}$. These gradient estimators closely follow the bias-variance tradeoffs as described previously. Specifically, we define
>
> $$
> \hat{\text{grad}}\_{SFT} = \text{grad}\_{oracle}(\theta) + \epsilon\_{variance} + \epsilon\_{bias}
> $$
>
> $$
> \hat{\text{grad}}\_{GRPO} = \text{grad}\_{oracle}(\theta) + 0.1 \cdot \epsilon\_{variance} + 0.1 \cdot \|\|\text{grad}\_{oracle}(\theta)\|\| \cdot \epsilon\_{bias}
> $$
>
> In each calculation, bias uses a deterministic variable that is predefined, and variance follows an iid normal distribution. Notably, we use grad norm to denote the distance between the current iterate $\theta$ and the optimal solution $\theta^*$. The results are shown in: <https://postimg.cc/xXfL537K> with code available in: <https://anonymous.4open.science/r/HPT-Re-F81B/HPT%20Case.ipynb>
>
> It can be observed that the SFT-style gradient estimation converges significantly faster at first; however, as the training process continues, the high variance dominates the gradient update, preventing $\theta$ from further decreasing. On the other hand, the GRPO-style gradient estimation converges more slowly at the start, but it gradually outperforms SFT-style gradient estimation as $\theta$ gets closer to $\theta^*$.
>
> In this case, we can derive an HPT-style algorithm by dynamically adjusting from the two gradient estimators: when $\|\|\text{grad}\_{oracle}(\theta)\|\|$ is large, use the SFT-style gradient; when $\|\|\text{grad}\_{oracle}(\theta)\|\|$ is small, use the GRPO-style gradient. It is clear that HPT outperforms both SFT and GRPO-style updates in this numerical example.

---

### Author Response · Authors · 2025-12-02
**Summary of Rebuttal**

Dear Area Chair,

Thank you sincerely for your careful consideration. Almost all reviewers mentioned that our work is well written, easy-to-follow, and well motivated. The reviewers also raised some questions and comments, to which we have mainly provided the following responses and supplements:

1. **More out-of-distribution tasks** (Reviewer PwLU). We additionally evaluate on three out-of-distribution tasks to assess the generalization ability of HPT. HPT achieves the comparative or best performance on all five benchmarks, which demonstrates that it leads to better generalization on OOD tasks.

2. **Computational cost analysis** (Reviewer PwLU). We analyze the computational costs of HPT and SFT→GRPO methods. And experimentally show that, under comparable GPU memory usage, the training time of SFT→GRPO is significantly higher than that of HPT.

3. **Connection between theoretical motivations and HPT** (Reviewers tTEQ & HJxf). We provide a more detailed explanation and deeper discussion in the General Response, including: (i) the connections between the Common Objective and classical RL literature, (ii) the source of bias and variance in LLM post-training, and (iii) the connection between UPGE and HPT: why HPT selects a better bias-variance tradeoff with a numerical optimization example.

4. **Statistical significance** (Reviewer HJxf). We perform repeated experiments from both the evaluation and training stages with different random seeds. The resulting means and standard deviations further support the effectiveness of HPT.

5. **More backbone models** (Reviewer dbLE). We additionally conducted experiments on Deepseek-Math-7B-base. HPT still achieves the best performance on Deepseek-Math-7B-Base, which further demonstrates the robustness of our method across different backbone models.

6. **Guidance on gate value setting** (Reviewers dbLE & t8ra). We provide practical guidance on how to set the gate value and further show that the gate threshold is not a sensitive component of HPT. Although tuning the gate is heuristic, our method guarantees that performance never decreases relative to the baseline.

We sincerely appreciate your efforts during this challenging period and are grateful for your careful consideration of our work.

---

### Note · Program_Chairs · 2026-01-17
**Submission Desk Rejected by Program Chairs**

The following references in this submission do not refer to real documents and/or have major errors in bibliographic information:

     Chunting Zhou, Graham Neubig, and Junxian He. Prefix-tuning for guided text generation in reinforcement learning. Transactions of the Association for Computational Linguistics, 11:12341249, 2023.
    Min-Joon Kim, Aviral Singh, and Hong-Seok Lee. Dynamic policy fusion for mixed-signal llm alignment. In Third Conference on Language Modeling, 2025.
    Jia Li, Zhaofeng Wang, and Junxian He. Self-guided exploration with offline demonstrations for complex reasoning. In Proceedings of the International Conference on Learning Representations, 2025.
    Jason Liu, Zhiyuan Chen, and Ji-Woo Park. Direct fine-tuning on rewarded trajectories for language model alignment. arXiv preprint arXiv:2406.13581, 2024.
    Zihan Liu, Alekh Agarwal, and Nan Jiang. A principled analysis of offline preference optimization algorithms. Journal of Machine Learning Research, 26(45):1-58, 2025c.
    Hao Liu, Zixuan Ji, and Di Lu. Bridging the gap between supervised fine-tuning and reinforcement learning. arXiv preprint arXiv:2308.08809, 2023.
    Ji-Woo Park, Yifan Chen, and Denny Zhou. Reward-reweighted sft: An offline policy refinement method. arXiv preprint arXiv:2502.11842, 2025.
    Jeff Wu, Long Ouyang, and Nisan Stiennon. Alternating between on-policy and off-policy updates for efficient and stable llm alignment. arXiv preprint arXiv:2401.08543, 2024.
    Hugo Touvron, Louis Martin, and Guillaume Lample. Context distillation for on-policy reinforcement learning in llms. In First Conference on Language Modeling, 2024.